
# The Regional Ice Ocean Prediction System v2: a pan-Canadian ocean analysis system

Gregory C. Smith[1], Yimin Liu[1], Mounir Benkiran[3], Kamel Chikhar[2], Dorina Surcel Colan[2], Charles-Emmanuel Testut[3], Frederic Dupont[2], Ji Lei[2], François Roy[1], Jean-Francois Lemieux[1] , and Fraser Davidson[4]

[1]Meteorological Research Division, Environment and Climate Change Canada (ECCC), Dorval, H9P1J3, Canada

[2]Meteorological Service of Canada, ECCC, Dorval, H9P1J3, Canada

[3]Mercator Océan International, Toulouse, France

[4]Northwest Atlantic Fisheries Centre, Fisheries and Ocean Canada, St. John's, Newfoundland, Canada

*Correspondence to*: Gregory C. Smith (Gregory.Smith2@canada.com)

**Abstract.**

Canada has the longest coastline in the world and includes a diversity of ocean environments, from the frozen waters of the Canadian Arctic Archipelago to the confluence region of Labrador and Gulf Stream waters on the East Coast. There is a strong need for a pan-Canadian operational regional ocean prediction capacity covering all Canadian coastal areas, in support of marine activities including emergency response, search and rescue as well as safe navigation in ice-infested waters. Here we present the first pan-Canadian operational regional ocean analysis system developed as part of the Regional Ice Ocean Prediction System version 2 (RIOPSv2) running in operations at the Canadian Centre for Meteorological and Environmental Prediction (CCMEP). The RIOPSv2 domain extends from 26°N in the Atlantic Ocean through the Arctic Ocean to 44° N in the Pacific Ocean, with a model grid-resolution that varies between 3 and 8 km. RIOPSv2 includes a multi-variate data assimilation system based on a reduced-order extended Kalman filter together with a 3DVar bias correction system for water mass properties. The analysis system assimilates satellite observations of sea level anomaly and sea surface temperature, as well as in situ temperature and salinity measurements. Background model error is specified in terms of seasonally varying model anomalies from a 10-year forced model integration allowing inhomogeneous anisotropic multi-variate error covariances. A novel online tidal harmonic analysis method is introduced that uses a sliding-window approach to reduce numerical costs and to allow time-varying harmonic constants, necessary in seasonally ice-infested waters. As compared to the Global Ice Ocean Prediction System (GIOPS) running at CCMEP, RIOPSv2 also includes a spatial filtering of model fields as part of the observation operator for sea surface temperature. In addition to the tidal harmonic analysis, the observation operator for sea level anomaly is also modified to remove the inverse barometer effect due to the application of atmospheric pressure forcing fields. RIOPSv2 is compared to GIOPS and shown to provide similar innovation statistics over a 3-year evaluation period.





Specific improvements are found in the vicinity of the Gulf Stream for all model fields due to the higher model grid-resolution, with smaller root-mean-squared (RMS) innovations for RIOPSv2 of about 5 cm for SLA and 0.5°C for SST. Verification against along-track satellite observations demonstrates the improved representation of meso-scale features in RIOPSv2

compared to GIOPS, with increased correlations of SLA (0.83 compared to 0.73) and reduced RMS differences (12 cm compared to 14 cm). While the RIOPSv2 grid resolution is 3 times higher than GIOPS, the power spectral density of surface kinetic energy provides an indication that the effective resolution of RIOPSv2 is roughly double that of the global system (35 km as compared to 66 km). Observations made as part of the Year of Polar Prediction (2017-19) provide a rare glimpse at errors in Arctic water mass properties and show salinity biases of 0.3-0.4 psu in the eastern Beaufort Sea in RIOPSv2.

**1. Introduction**

Over recent years, there has been a growing number of regional ice-ocean prediction systems developed and being run operationally (Kourafalou et al., 2015; Tonani et al., 2015). For example, as part of the European Copernicus Marine Environmental Monitoring Service (CMEMS) there are regional systems covering the Arctic Ocean (Sakov et al., 2012), the European Northwest Shelf Seas (King et al., 2018), the Iberia-Biscay-Ireland Shelf Seas (Sotillo et al., 2015), the

Mediterranean Sea (Tonani et al., 2008) and the Baltic Sea (Zhuang et al., 2011). Various systems are also in place along the US coastlines (e.g. Zhang et al., 2010; Moore et al., 2011; Xue et al., 2005; Mehra and Rivin, 2010). More recently, a number of systems have been put in place by China (Cho et al., 2014), Japan (Hirose et al., 2019) and Korea (Park et al., 2015).

Developing a regional ocean prediction system for Canada is challenging due to the length of the Canadian coastline and

complexity of coastal waters in Canada. First, Canada's coasts cover three oceans that differ greatly: from relatively warm North Pacific Ocean waters on the west coast; the confluence region of the Gulf Stream and the Labrador Current on the east coast; to seasonal and multi-year sea ice in the Canadian Arctic Archipelago. Moreover, all three regions experience strong tidal currents, including the two largest tidal ranges in the world in the Bay of Fundy and Ungava Bay (Arbic et al., 2007; O`Reilly et al., 2005). A paucity of in situ data further complicates the development of reliable data assimilative operational

oceanographic systems, due in particular to significant gaps in real-time data availability. This situation is especially challenging in the Canadian Arctic Archipelago due to the harsh conditions that are present, combined with a dynamic sea ice cover and further complicated by significant regions of poorly known bathymetry.

Early efforts in operational oceanography in Canada include models run in real-time without any data assimilation to support

search and rescue and oil spill response for the Gulf of St. Lawrence (Saucier et al., 2003) and St. Lawrence Estuary (Saucier and Chassé, 2000). The Gulf of St. Lawrence model was later developed into a coupled prediction system for sea ice and weather prediction (Pellerin et al., 2004; Smith et al., 2012). Additionally, a coupled ice-ocean prediction system for the Canadian East Coast used for iceberg drift, ice service operations (i.e. for safe navigation) and offshore resource exploitation



was implemented in 2007 that assimilates sea surface temperature and ice concentration (Tang et al., 2008). On the Canadian
west coast there have been numerous modelling efforts (e.g. Masson and Cummins, 2007), although few real-time forecasting
efforts, apart from tidal and storm surge forecasting systems (e.g. Soontiens et al., 2016).

More recently, the Global Ice Ocean Prediction System (GIOPS) was implemented at the Canadian Centre for Meteorological
and Environmental Prediction (CCMEP) providing the first operational global ocean assimilative capacity in Canada (Smith
et al., 2015). This system was designed primarily to support the initialization of coupled medium-range deterministic weather
prediction (Smith et al., 2018). It is also now used to initialize coupled ensemble predictions on sub-seasonal (medium-
monthly) range  as well as for seasonal predictions (Lin et al., 2020). GIOPS analyses and forecasts are also used by the
Canadian navy for marine operations (e.g. sonar range prediction). Building on the availability of ocean analyses from GIOPS,
a higher grid-resolution Regional Ice Ocean Prediction System (RIOPS) was developed (Dupont et al., 2015; Lemieux et al.,
2016a) to support marine operations in Canadian ice-infested waters, and in particular, over two Arctic METAREAs (17 &
18) for which Canada has the responsibility to provide warnings regarding ice hazards and marine weather predictions as part
of the Global Marine Distress and Safety System.

Increasing requirements for a world-class safety system to protect Canadian coastal areas has motived the creation of the
Government of Canada Ocean Protection Plan initiative. This initiative aims to put in place a range of measures, including a
pan-Canadian ocean prediction capacity to provide numerical guidance for marine emergency response (e.g. oil spill). To meet
this need, ocean predictions are required that are able to represent coastal ocean processes (e.g. tidal flows, boundary currents)
while constraining internal variability (i.e. through data assimilation).

There has yet to be a pan-Canadian regional ocean prediction system capable to meet this need. Here we present the first such
system in RIOPSv2, implemented operationally at the CCMEP on July 3rd, 2019 and developed as part of the Canadian
Operational Network for Coupled Environmental Prediction Systems (CONCEPTS) initiative (Smith et al., 2013). RIOPSv2
includes both an extended domain to cover the North Pacific Ocean (Fig. 1) as well as a multi-variate ocean data assimilation
system. The ocean data assimilation component is similar to that used in the GIOPS system with several important additions.
First, the sea level anomaly observation operator must be modified to filter tidal variations.  Given the seasonal variations in
tidal harmonics due to the presence of sea ice, a novel online harmonic analysis with a sliding-window approach is introduced.
Second, a spatial filter is added to the sea surface temperature (SST) observation operator to remove small-scale features not
resolved in the SST analysis product assimilated. The background error is defined in terms of a set of error modes derived
from a multi-year forced simulation following the approach described by Lellouche et al. (2013). The Incremental Analysis
Updating (IAU) period is extended from 1 day used by GIOPS to 7 days to provide improved continuity. As a result, the
RIOPSv2 system has many aspects in common with the CMEMS global prediction system (Lellouche et al.; 2013). Particular



differences include the use of a regional domain, the inclusion of tidal SSH variations, a 3DVar sea ice analysis, the assimilation of the CCMEP SST and the use of a daily update analysis using a 1-day assimilation window used to initialize forecasts.

Here we present the RIOPSv2 system and provide an evaluation and demonstration of the added-value with respect to the GIOPS analysis system. Section 2 provides a detailed description of the ocean and sea ice numerical models and describes improvements with respect to the previous RIOPSv1.3 system. Section 3 presents an overview of the assimilation components and details the various modifications made for RIOPSv2 including the new online harmonic analysis method introduced here. Section 4 presents an evaluation of innovations of sea level anomaly, SST, and temperature and salinity profiles over a 3-year

reanalysis period. A comparison of RIOPSv2 and GIOPS along a particular Jason satellite altimeter track that crosses the Gulf Stream is also provided in Section 4 as well as an analysis of the power spectral density of the surface kinetic energy fields of RIOPSv2 and GIOPS. Conclusions and a discussion of future work are provided in Section 4.

The main contributions of this paper are : a description of the first pan-Canadian regional ocean analysis system; the

development of a novel online harmonic analysis method with a sliding window approach; and a demonstration of the added value of the RIOPSv2 analysis system compared to GIOPS in terms of smaller innovations over the Gulf Stream and higher effective resolution.

## 2. Numerical model description

### 2.1 Ocean model component

The ocean model used in RIOPSv2 and GIOPS is the Océan Parallisé (OPA) model as part of the Nucleus for European Modelling of the Ocean (NEMO; Madec et al., 2008) modelling framework. OPA is a primitive equation model on an Arakawa 'C' grid  employing the non-Boussinesq and hydrostatic approximations. A detailed description and evaluation of the model is provided in Dupont et al. (2015). The particular parameterization details and settings used are described in Tables 1 and 2.

The RIOPSv2 numerical grid is constructed from the 1/12° tri-polar 'ORCA' grid (Madec and Imbard, 1996), whereby grid-points from the Pacific section have been re-mapped to the top of the grid to eliminate the north fold from the ORCA grid (See Dupont et al., 2015 for details). The resulting grid has been previously referred to as the CONCEPTS Regional grid (CREG) as used in various studies (e.g. Roy et al., 2015; Chikhar et al., 2019; Boutin et al., 2020). In RIOPSv2 the CREG grid is extended to include the North Pacific Ocean down to 44°N in order to cover the Canadian west coast. Bathymetry is based on

ETOPO2 (Amante and Eakins, 2009). Tidal variations in sea surface height (SSH) and barotropic transport are applied along the open boundaries in the Atlantic and Pacific Oceans using 13 tidal constituents (M2, S2, N2, K2, O1, K1, Q1, P1, M4, Mf, Mm, Mn4, Ms4) extracted from the Oregon State University product (Egbert and Erofeeva, 2002). Self-attraction and loading terms are prescribed following the Finite Element Solution (FES) 2012 tidal product (Carrère et al., 2012). An evaluation of

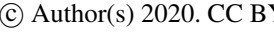



tidal variations in the CREG configuration is presented in Lemieux et al. (2018) and regional evaluations for the 1/12°
resolution model configuration used in RIOPSv2 can be found in Nudds et al. (2020) and Paquin et al. (in prep.).

In RIOPSv2, the ocean model component was updated to NEMOv3.6 (from v3.1 with various modifications). As many features
from NEMOv3.6 had been back-integrated in the NEMO code version used in RIOPSv1.3, this change in code did not in itself
provide any significant change to model results. Rather, the code was updated to permit use of the XIOS I/O server.


Several other changes were made to the ocean model configuration (Table 1). The number of vertical levels was increased
from 50 to 75 levels, in order to improve the resolution of the deep layers (from 500 m to 250 m). Atmospheric pressure forcing
is added such that the atmospheric pressure gradient may be applied in the momentum equations (the so-called "inverse
barometer" effect) as in storm surge modelling. Following stability tests, the time-step was increased to 300 s. The vertical
background diffusivity and viscosity were reduced respectively to molecular values ($10^{-7}$ and $10^{-6}$ m$^2$ s$^{-1}$) to reduce mixing in
deeper water masses in the Arctic.

## 2.2 Sea ice model component

RIOPSv2 uses the Los Alamos Community Ice CodE version 4.0 (CICE; Hunke, 2001; Lipscomb et al., 2007; Hunke
and Lipscomb, 2008). CICE is a dynamic-thermodynamic sea ice model on an Arakawa B-grid. Ice-ice interactions are
governed using an Elasto-Viscous Plastic scheme (Hunke et al., 2001). In RIOPS, CICE is configured to use 10 ice thickness
categories and a single snow layer. Landfast ice is parameterized using the basal-stress approach of Lemieux et al. (2015,
2016b).

In RIOPSv2 the model code and settings were kept identical with the exception of the atmosphere-ice and ice-ocean drag
coefficients (Table 1). In an attempt to reduce the existing negative bias in ice velocity, the ice-ocean roughness was reduced
from 2.6 cm to 2.0 cm, and the air-ice roughness increased from 0.16 mm to 0.20 mm. In multi-annual hindcast simulations
produced over the period 2003-2008 this change was found to reduce the negative bias in ice drift with respect to buoys from
the International Arctic Buoy Program (Rigor and Ortmeyer, 2004) as well as the gridded drift product from the National Snow
and Ice Data Centre (Tschudi et al., 2019; not shown).

## 3. Data assimilation methods

In this section, we present the data assimilation systems for the ocean and sea ice. The main innovation introduced here is the
online harmonic analysis method with a sliding-window approach described in Section 3.4. The sea ice concentration and sea
surface temperature analyses used by RIOPSv2 are independent of (i.e. not cycled with) the ocean data assimilation system
and only a brief description is provided. Note that an update to the version of RIOPS running operationally is expected in fall





2021 to include a 3DVar bias correction as is used for GIOPSv3. The specific version of RIOPS described here (v2.1) includes this addition (referred to hereafter simply as RIOPSv2).

In place of the spectral nudging approach used to generate initial conditions for RIOPSv1, in RIOPSv2 a multivariate data assimilation approach is implemented. This system is based on the System d'Assimilation Mercator version 2 (SAM2;

Lellouche et al., 2013) system used for GIOPS (Smith et al., 2015; 2019a). This system assimilates satellite observations of sea level anomaly and sea surface temperature, together with in situ observations of temperature and salinity. Ongoing evaluations (e.g. Zedel et al., 2018) and intercomparison activities as part of GODAE Oceanview (Bell et al., 2015) have shown GIOPS to provide analyses and forecasts of similar skill to other operational global ocean forecasting systems (Ryan et al., 2015; Divakaran et al., 2015). Following the approach used in GIOPS, the RIOPS system has three assimilation cycles

(Fig. 2): a daily update (assimilating only SST and sea ice concentration with a 1-day analysis window) called 'RU', a real-time weekly cycle 'RR' and a delayed weekly cycle 'RD' run 7 days behind real-time. The RD cycle is the backbone of the system and provides the continuity in time. The evaluation presented here uses RD cycles.

As the GIOPSv3 system will be used as a reference for the evaluation (Section 4), differences between RIOPSv2 and GIOPSv3

are highlighted in the following sub-sections.

### 3.1 Overview of the ocean data assimilation system

The SAM2 ocean data assimilation system is a reduced-order extended Kalman filter using a Singular Evolutive Extended Kalman (SEEK) Filter methodology (Pham et al., 1998). The main properties and features of the scheme are as described in Lellouche et al. (2013; 2018). A brief description is provided below. In the following section, specific adaptations made in the

GIOPS system (Smith et al., 2015) and used for RIOPSv2 are described.

The background error is defined in terms of a static set of multi-variate error modes obtained from sub-monthly anomalies of a multi-year forced simulation. A description of the method used for the construction of these error modes is provided in Section 3.3.2. An adaptivity scheme based on Talagrand (1998) is applied to adjust model background error variances.

Innovations are calculated during the model integration in a First Guess at Appropriate Time (FGAT) approach. Two online quality-control checks are applied to in situ temperature and salinity profiles based on temperature and salinity innovations and dynamic height innovations. Analysis increments are applied gradually in an Incremental Analysis Updating (IAU) approach (Bloom et al., 1996). A 3DVar bias correction approach is used for temperature and salinity profiles using mean innovations from the previous 4 cycles.

In order to assess satellite observations of sea level anomaly in the observation operator, a Mean Dynamic Topography (MDT) must be first removed from the model sea surface height. As a result, errors (or inconsistencies with the model) in the mean



dynamic height field used may result in systematic errors in analysis increments and persistent mean increments in some regions. Here, the hybrid product described by Lellouche et al. (2018) is used. This product combines the CNES-CLS13 MDT (Rio et al., 2014) with mean increments calculated from the Mercator Ocean GLORYS2V3 reanalysis.

Several features described in Lellouche et al. (2018; system referred to therein as PSY4V3R1) are not used here. These include a Desroziers et al. (2005) scheme to adjust observation error variances and the application of a weak constraint toward climatology in the deep ocean. An additional difference is that the sea ice assimilation is not done using SAM2 but rather with a separate 3DVar approach (described in Section 3.2 below).

## 3.2 Adaptations of SAM2 for GIOPS

A number of modifications were made to SAM2 for use in the GIOPS system and are described in Smith et al. (2015a,b). First, the CCMEP gridded SST analysis (Brasnett and Colan, 2016) is assimilated (as opposed to the OSTIA or AVHRR products used in Lellouche et al. (2018)). This foundation SST analysis is produced on a 0.1° resolution latitude-longitude grid using an Optimal Interpolation (OI) approach that assimilates AMSR-2, AVHRR, VIIRS, and in situ observations from moorings,

ships and drifters. The OI uses the previous day's analysis as first guess for the present day's analysis. The SST OI analysis is set to the model freezing temperature in locations where the ice concentration is greater than 0.6. Additionally, SST observations within one grid-cell of the coastline are rejected. The SST OI analysis is then assimilated using a relatively small 0.3°C observation error. This provides a tight constraint to the SST analysis necessary to reduce initialization shock for coupled forecasts (Smith et al., 2018). Another modification required for coupled forecasts was to use 24-h averaged short and long-

wave radiation fields to force NEMO-CICE during the analysis cycles such that there is very little diurnal warming present in the ocean analysis..

At the end of each analysis cycle, the ocean analysis is blended with a 3DVar sea ice concentration analysis (Buehner et al., 2013, 2016). This blending adjusts the concentration of the 10 ice thickness categories using the Rescaled Forecast Tendency

(RFT) method of Smith et al. (2015). The 3DVar ice analyses are produced on a 10-km grid and assimilate passive microwave retrievals from SSMI and SSMI/S using the NasaTeam 2 algorithm, AMSR2, ASCAT as well as manual Radarsat image analyses and ice charts from the Canadian Ice Service.

## 3.3 Modifications introduced for RIOPSv2

The SAM2 system used in GIOPS had to be adapted for use in a regional context. As a result, there are several important

differences in the assimilation approach used in RIOPSv2 described in the following sections. The most notable change is the introduction of the online harmonic analysis that will be described separately in the following section (Section 3.4).





### 3.3.1 7-Day Incremental Analysis Updating

In the GIOPS system, analysis increments are applied using an Incremental Analysis Updating (IAU; Bloom et al., 1996) approach with increments applied evenly over a 1-day period. For the delayed-mode (GD) and real-time (GR) weekly cycles

this is done over the last day of the 7-day assimilation window and for the daily cycle (GU) this is done over the full 1-day window. To provide greater consistency in time, in RIOPSv2 a 7-day IAU is used for both RD and RR cycles inspired by the methodology of Benkiran and Greiner (2008). A linearly-increasing ramp is used over the first day, followed by a constant increment for days 2-7. A decreasing ramp is applied for the first day of the following cycle. This approach is also applied in systems used by Mercator Océan International (Lellouche et al., 2013; 2018).


### 3.3.2 Background error

As mentioned above, the background error for RIOPSv2 is specified in terms of a series of error modes derived from a multi-year forced simulation over the period 2002-2011. Sub-monthly anomalies are constructed by removing low-frequency variations using a 30-day Hanning Filter. The basic methodology used to calculate the error modes is the same as used for

GIOPS apart from the smoothing used in the error modes. For RIOPSv2, 49 passes of a Shapiro filter are applied to temperature and salinity fields resulting in a roughly 1° smoothing. Also, tides are removed from the SSH field using the online harmonic analysis described below prior to calculation of the sea surface height anomaly.

The resulting error modes provide a seasonally-varying estimate of the background error. For a particular analysis cycle, error

modes within a 90-day window around the Julian day of the analysis cycle from all 10 years of the forced simulation are used. As error modes are produced every 3 days, this provides roughly 300 error modes available for each analysis cycle.

While this approach does not provide an estimate of "errors of the day" as in an ensemble Kalman Filter, it does nonetheless provide an estimate of the climatological covariances both in space and between model variables (i.e. SSH and three-

dimensional fields of temperature, salinity, and zonal and meridional velocities). An example of the multi-variate correlations are shown in Figs. 3-6. Correlations can vary considerably at different locations and between variables based on the underlying oceanographic conditions. For example, in the Gulf Stream (Fig. 3) strong correlations are present between SSH and temperature at short distances, representative of dynamic height variability in this region of strong mesoscale activity. Along the Labrador Shelf (Fig. 4), correlations of SST fields are strongly anisotropic and oriented along the axis of the Labrador

Current. Correlations between SST and sub-surface temperature are modulated by local (and seasonal) variations in mixed layer depth. In the Beaufort Sea (Figs. 5 and 6), we see pronounced differences between on-shore and off-shore correlations. On-shore we find larger-scale SST correlations across the shelf, whereas off-shore correlations are highly localized.





As a SEEK filter formulation is used, the model background error will determine the sub-space within which the analysis is
restricted. Here, this sub-space is limited by the variability of the model fields over the 10-year period of the model simulation.
As such, the spatial scales represented in the analysis are determined by the effective model resolution and the spatial filtering
applied to the error modes.

### 3.3.3 SSH observation operator

To provide the most accurate model equivalent possible, it is important to apply the same processing to model fields as has
been done to satellite altimeter observations (e.g. Carrère et al., 2003; Dibarboure et al., 2011). As RIOPS includes tidal and
atmospheric pressure forcing, the SSH observation operator must also be adjusted to filter these sources of variability as they
have been filtered from the AVISO Salto/Duacs SLA observations that are assimilated. The inverse barometer effect can be
calculated locally based on the hourly atmospheric pressure forcing applied to the model and accounted for directly. This will
not capture non-local effects such as coastally-trapped waves. As the SSH observation operator applies a 24-h mean, much of
the remaining coastal variability will be removed. Moreover, as the SLA observations are assimilated with a larger error close
to coasts these effects will not have a significant impact. The SSH observation operator is also modified to include the online
harmonic analysis as described in Section 3.4.

### 3.3.4 SST observation operator

As noted above, for both RIOPSv2 and GIOPSv3 the SST observations are assimilated in the SAM2 system using the CCMEP
SST OI analysis. As this OI analysis uses covariances with e-folding length scales varying between 20-70 km (Brasnett and
Colan, 2016) it is appropriate to apply a spatial filter to model SST fields to match the spatial scales present in the OI analysis.
A Shapiro filter with 49 passes is used in RIOPS to provide a roughly 1° resolution field. This number of passes was chosen
as it was found to provide a good match in the power spectral density of SST fields between the CCMEP analysis and RIOPS
fields (not shown). Moreover, as a diagonal observation error covariance matrix is used in SAM2, the CCMEP analysis is
decimated by 1 point out of 5 to reduce correlated variability. Figure 8 provides an example for 20-Jul-2016 showing the
unfiltered 0.1° resolution CCMEP SST analysis, the CCMEP analysis decimated to 0.5° resolution, the model trial field for
the same date and the model equivalent following application of the Shapiro filter. We can clearly see that the raw model fields
(Fig. 7c) contain smaller scales than the CCMEP analysis, whereas the filtered model fields provide a more representative
comparison to the CCEMP analysis. The innovation is then calculated as Fig. 7 (b) minus (d).




### 3.4 Online harmonic analysis

Removing tidal variability requires a careful approach as any tidal-variations not filtered will contribute directly to the
innovations and may result in unphysical increments. A common approach is to perform a harmonic analysis of the primary
tidal constituents over a model simulation of an appropriate length (Foreman and Henry, 1989) to adequately differentiate
between constituents of similar frequency (e.g. a month for diurnal and semi-diurnal constituents and a year for fortnightly
and monthly constituents). However, this approach neglects non-stationary effects such as seasonally-varying interactions
between barotropic and baroclinic tidal modes. Moreover, in areas of sea ice cover, a strong seasonal cycle in tidal harmonics
may be present. Indeed, Kleptsova and Pietrzak (2018) show summer-winter differences in the M2 amplitude that exceed 1 m
in Hudson Strait and Ungava Bay, with changes in phase up to 180° due to the displacement of tidal amphidromes.

Here we introduce a new online harmonic analysis method that uses a sliding-window approach to update harmonic
coefficients. This approach provides an accurate estimate of tidal variability with a relatively low computational cost. Our
scheme is similar to other harmonic analysis approaches (e.g. T-Tide; Pawlowicz et al., 2002) in that it is based on a least-
square fit of the SSH time series by a set of harmonic functions. Here we adapt this method for efficient online model
computation using a sliding-window approach with a given time weight to allow harmonic coefficients to vary in time. A
detailed description of the method is given below including a derivation of the basic harmonic analysis equations (Section
3.4.1), the use of a rotation operator for the sliding window approach (Section 3.4.2). A discussion of the numerical advantages
of the sliding window approach are presented in Section 3.4.3. Finally, the implementation details are provided in Section
3.4.4, together with an example showing the results of the method. The derivation of the real-space equations implemented in
NEMO for RIOPSv2 are provided in Appendix A.

### 3.4.1 Basic description

Given the model SSH time series $H^n$ at the $n^{th}$ model time step at a particular point on the model 2-dimensional grid, we aim
to find the harmonic-spectrum coefficient $X^k$ for each $k^{th}$ frequency to provide the harmonic decomposition

$$A^n = E_k^n X^k = E_k^{*n} X^{*k} , \tag{1}$$

where superscript * indicates complex conjugate, and $A^n$ is the decomposition of the real time series $H^n$. The harmonic function
base is defined as

$$E_k^n = C exp(i\omega_k \tau n) , \tag{2}$$

where $C$ is a time-independent normalization constant factor, which could include the frequency dependent initial phase term,
$\omega_k$ at the $k^{th}$ pre-chosen harmonic frequency, and $\tau$ is the harmonic analysis time step length. Note that here we use $n$ and $m$
as indexes of time step and $k$ and $j$ as the frequency index, $k=0,1,2, ..., K$. In the RIOPS system, we select $K=33$, and specify
$\omega_0=0$ for the mean of time series $H$.





$A^n$ is found by minimizing the following cost function

$$J = \tfrac{1}{2}(A^n - H^n)W_{nm}(A^m - H^m) = \tfrac{1}{2}(E^n_k X^k - H^n)W_{nm}(E^m_j X^j - H^m) \ , \qquad (3a)$$

$$= \tfrac{1}{2}(E^n_k X^k - H^n)^* W_{nm}(E^m_j X^j - H^m), \qquad (3b)$$

where $\boldsymbol{W_{nm}}$ is a real diagonal matrix of the time weights used in the sliding window; the indexes $n$ and $m$ take values from **-**
$N+1,...,-2,-1,0$, where $n=0$ indicates the current step. $N$ is the sliding window length in units of model time-steps and increases

with model integration time. Hereafter, we follow the Einstein Summation Convention, i.e. summing over the covariant and
contravariant index pairs, unless noted otherwise.

According to Eq. (3b), the variation of $\boldsymbol{J}$ on $X^{*k}$ has

$$\frac{\delta J}{\delta X^{*k}} = E^{*n}_k W_{nm}(E^m_j X^j - H^m) = (E^{*n}_k W_{nm} E^m_j)X^j - E^{*n}_k W_{nm}H^m \ , \qquad (4)$$

We denote $\boldsymbol{B_{kj}} = \boldsymbol{E^{*n}_k W_{nm} E^m_j}$ with size $(K+1) \times (K+1)$, and $\boldsymbol{Y_k} = \boldsymbol{E^{*n}_k W_{nm} H^m}$ with size $(K+1) \times 1$. If we set Eq. (4) to be equal
to zero, we can obtain the minimization solution of $\boldsymbol{J}$ as following

$$X^j = (B^{-1})^j_k Y^k \ , \qquad (5)$$

where, $\boldsymbol{B^{-1}}$ is the inverse matrix of $\boldsymbol{B}$. Hence, we can obtain the final solution of the tidal harmonic analysis as

$$A^n = E^n_j(B^{-1})^j_k Y^k \ . \qquad (6a)$$

Note that the diagonal matrix $\boldsymbol{W}$ guarantees that matrix $\boldsymbol{B}$ is Hermitian and $A^n$ is real. From Eq. (6a) we can see that the constant
$C$ in $\boldsymbol{E^n_k}$ (Eq. 2) can be cancelled, which means the final solution $\boldsymbol{A}$ is independent of $C$. Therefore, we set $C=1$ without any
loss of generality. A convenient feature of this approach when implemented as an online harmonic analysis in a numerical
model is that we only need the value of $\boldsymbol{A^0}$ from the current time step. Following Eq. (2), $\boldsymbol{E^0_j}$ is equal to 1, so that we have

$$A^0 = \sum_{j=0}^{K}[(B^{-1})^j_k Y^k]. \qquad (6b)$$

Note that the mean of time series $\boldsymbol{H}$ is included in the above $\boldsymbol{A^0}$.

### 3.4.2 Sliding window approach

In the following, we will show how to use the rotation operator and a sliding window approach to update the $\boldsymbol{B}$ and $\boldsymbol{Y}$ matrices
for the current time step based on previous time steps $\boldsymbol{B'}$ and $\boldsymbol{Y'}$; hereafter, the symbol ' indicates the previous time step.
First, we split the sum index range of $n$ (and $m$) into two sets, one for current time step and containing only step index $\boldsymbol{0}$, and

another set for previous steps containing indexes $-N+1, ..., -2, -1$; to differentiate, we will use indices $p$ and $q$ for the latter set.
Therefore, considering that $\boldsymbol{W}$ is diagonal matrix, we can rewrite the $\boldsymbol{B}$ and $\boldsymbol{Y}$ matrices as follows:

$$B_{kj} = E^{*n}_k W_{nm} E^m_j = E^{*0}_k W_{00} E^0_j + E^{*p}_k W_{pq} E^q_j \ , \qquad (7a)$$

$$Y_k = E^{*n}_k W_{nm} H^m = E^{*0}_k W_{00} H^0 + E^{*p}_k W_{pq} H^q \ . \qquad (7b)$$

According to Eq. (2), $\boldsymbol{E^0_k}$ and $\boldsymbol{E^{*0}_k}$ are simply equal to 1. If we denote $\boldsymbol{W_{00}}$ as $\alpha$ , which is specified as the ratio of time step

length $\tau$ to a given restoring time length (e.g. 30 days in RIOPSv2),  then the first terms on right-hand sides (RHS) of Eq.





(7a,b) become $\alpha 1_{kj}$ and $\alpha V_k H^0$; here $1_{kj}$ and $V_k$ are a matrix and a vector with all elements equal to real number 1, respectively.

In the second term on the RHS of Eq. (7b), we use $n$ (or $m$) to replace $p+1$ (or $q+1$), and denote $\boldsymbol{H^q = H'^m}$. Here, $\boldsymbol{H'^m}$ is the
variable $\boldsymbol{H}$ at the $m^{th}$ time step referred to in the previous step rather than to the current step. According to the definition of the sliding window weight and considering its property of normalization, the sliding window weight $\boldsymbol{W}$ can be specified as following:

$$W_{-1,-1} = (1-\alpha)\,W_{00}\,,$$

$$W_{-2,-2} = (1-\alpha)\,W_{-1,-1},\ \dots$$

$$W_{n-1,m-1} = (1-\alpha)\,W_{nm}\,.$$

In other words, $\boldsymbol{W_{pq}} = (1-\alpha)\boldsymbol{W_{nm}}$. As such, the second term on the RHS of Eq. (7b) can be rewritten as:

$$
\begin{aligned}
E_k^{*p} W_{pq} H^q &= \sum_{p,q=-N+1}^{-1} E_k^{*p} W_{pq} H^q \\
&= \exp(i\omega_k\tau) \sum_{p,q=-N+1}^{-1} \exp(-i\omega_k\tau) E_k^{*p} W_{pq} H^q \\
&= \exp(i\omega_k\tau) \sum_{p,q=-N+1}^{-1} E_k^{*p+1} W_{pq} H^q \\
&= \exp(i\omega_k\tau) \sum_{n,m=-N+2}^{0} E_k^{*n} (1-\alpha) W_{nm} H'^m \\
&= (1-\alpha) \exp(i\omega_k\tau) \sum_{n,m=-N+1}^{0} E_k^{*n} W_{nm} H'^m + O(W_{-N+1,-N+1}) \\
&= (1-\alpha) \exp(i\omega_k\tau) Y'_k + O(W_{-N+1,-N+1})\,. \qquad (8)
\end{aligned}
$$

Here, the term $O(W_{-N+1,-N+1})$ results from the change in the lower limit of the summation of $n(m)$ from $-N+2$ to $-N+1$. Generally speaking, this term can be neglected when $N$ is sufficiently large, because $W_{-N+1,-N+1}$ is very small, on the order
of $(1-\alpha)^{N-1}$ and $\alpha(1-\alpha)^{N-1}$ comparing with $\boldsymbol{W_{00}}$ and $\boldsymbol{Y}$, respectively. For example, in RIOPS with a one-month restoring time length, and after a half-year spin-up period, they are about $e^{-6} = 2.479 * 10^{-3}$ and $1/(288*30) * e^{-6} = 2.869 * 10^{-7}$, respectively.

Therefore, using Eq. (8) and Eq. (7b), we can obtain

$$Y_k = \alpha V_k H^0 + (1-\alpha) R_k^{*j} Y'_j\,, \qquad (9)$$

here, $\boldsymbol{R}$ is a diagonal time-independent rotation matrix, with its $k^{th}$ diagonal element taking the value

$$R_k^k = exp(-i\omega_k\tau)\,. \qquad (10)$$

The matrix $\boldsymbol{R}$ can be used to rotate a complex vector in the complex plane spanned by the $k^{th}$ complex harmonic function base. For instance, a complex vector $e^{i\theta}$ could be rotated into a vector $e^{i(\theta-\omega_k\tau)}$ by operator $\boldsymbol{R^k}_k$. One exception is that, when $\omega_0 =$
$0$, as is the case for the mean of time series of $H^n$, both $\boldsymbol{R^0_0}$ and $\boldsymbol{E^n_0}$ are equal to real number 1, so that, there is no complex plane spanned, and $\boldsymbol{R}$ only takes action in the 1-dimension real space.





Similarly, we can rewrite the RHS of Eq. (7a) as :

$$E_k^{*p} W_{pq} E_j^q = \sum_{p,q=-N+1}^{-1} E_k^{*p} W_{pq} E_j^q$$

$$= \exp(i\omega_k\tau)\exp(-i\omega_j\tau) \sum_{p,q=-N+1}^{-1} \exp(-i\omega_k\tau) E_k^{*p} W_{pq} E_j^q \exp(i\omega_j\tau)$$

$$= \exp(i\omega_k\tau)\exp(-i\omega_j\tau) \sum_{p,q=-N+1}^{-1} E_k^{*p+1} W_{pq} E_j^{q+1}$$

$$= \exp(i\omega_k\tau)\exp(-i\omega_j\tau) \sum_{n,m=-N+2}^{0} E_k^{*n} (1-\alpha) W_{nm} E_j^m$$

$$= (1-\alpha) \exp(i\omega_k\tau)\exp(-i\omega_j\tau) \sum_{n,m=-N+1}^{0} E_k^{*n} W_{nm} E_j^m + O(W_{-N+1,-N+1})$$

$$= (1-\alpha) \exp(i\omega_k\tau) B'_{kj} \exp(-i\omega_j\tau) + O(W_{-N+1,-N+1})$$

$$= (1-\alpha) R_k^{*i} B'_{il} R_j^l + O(W_{-N+1,-N+1}). \tag{11a}$$

Combining Eqs. (7a) and (11a) we can write

$$B_{kj} = \alpha 1_{kj} + (1-\alpha) R_k^{*i} B'_{il} R_j^l. \tag{11b}$$

Finally, based on Eqs. (6b) and (9-11b), we can get the final solution $A^0$ for the current time step. Appendix A provides the derivation of the equivalent real-space version of these equations that is coded in RIOPSv2.


### 3.4.3 Numerical advantages

The main advantage of this sliding window approach is that it is much cheaper to directly use equation $Y_k= E^{*n}_k W_{nm} H^m$ than the traditional method. For instance, in RIOPSv2 the number of horizontal grid points is $N_{xy}=1580\text{x}2198,$ and if we update tidal harmonics at each model step with a half-year spin-up-window (or analysis-time-window) implies a total number of time
steps of $N_{stp}=288\text{x}182$. This means the size of $Y$ is less than 2 Gigabyte ($\sim 8\text{x}N_{xy}\text{x}(2K+1)$) for our method and over 1 Terabyte ($\sim 8\text{x}N_{xy}\text{x}N_{stp}$) for the traditional method. In addition, to update $Y$ at each model time step using the traditional method by equation $Y_k= E^{*n}_k W_{nm} H^m$ would require the computation of sine and cosine decomposition operators $N_{xy}\text{x}(2K+1)\text{x } N_{stp}$ times. Alternatively, the method outlined here requires only the use of the multiplication operator $N_{xy}\text{x}(2K+1)$ times. Moreover, to speed up computation in the traditional method one could save the sine and cosine decomposed values at each model step,
however, the data size of $Y$ will become huge, over $(2K+1)$ Terabytes. As such, the traditional method would be too slow for operational systems based on the present-day computing environments, unless one would make some modifications and/or simplifications. For instance, the cost could be lowered by reducing the resolution of the temporal and the spatial calculations. However, this could have negative consequences in coastal areas and areas of steep bathymetric slope as the local geometric structures have a strong influence on the tidal harmonics, as do high-frequency tidal constituents.


As we assume that the matrix $B$ is homogeneous for the horizontal grids, their data size is very small ( $(K+1)\text{x}(K+1)$ in complex and $(2K+1)\text{x}(2K+1)$ in real-space, respectively), and there is no issue for $B$ regarding differences of computational costs, memory and I/O, between the traditional method and online harmonic analysis with the sliding window.





Another advantage of the sliding window approach is with respect to its spin-up. As we know, dynamic models and data assimilation systems require a certain time period to be spun-up to equilibrium. When applying the online tidal harmonic analyses, we can allow all three parts (model, assimilation and tidal filter) to be coupled in harmony, during the common spin-up time window from the model cold start.

**3.4.4 Implementation details**

This sliding window approach is implemented using the real-space version of the equations (Appendix A) such that Eqs. (A2-A4) are updated at every time step using 33 diurnal and semi-diurnal tidal constituents. Updating these equations less frequently (e.g. every 4 model time steps or hourly) was found to have a negative impact on the accuracy of the harmonic analysis and resulted in an increase in the tidal energy remaining in the SSH residual. The choice of 33 tidal constituents was based on the

results of an offline harmonic analysis made using T_tide (Pawlowicz et al., 2002). Additionally, no harmonics with a period longer than 30 h were used to avoid contamination of the tidal signal by mesoscale variability. For example, including the fortnightly (Msf) constituent resulted in a significant decrease in the eddy kinetic energy in the tidal residual. An extension of the method described here to take into account spatial correlations could avoid this limitation.

To illustrate the relative importance of the various filtering steps, the different components (tides, inverse barometer) have been separated and an example is shown in Fig. 8. The tidal signal is the most prominent source of variability and has a significant impact with local variations exceeding 1 m in amplitude. In comparison, the inverse barometer effect is generally below 0.2 m. The residual SSH with tides and inverse barometer removed shows well the main physical features including the Gulf Stream and Beaufort Gyre. Some tidal variability remains in coastal areas with strong tides (e.g. Bay of Fundy), but this

is expected to have a negligible effect as SLA observation errors are amplified near-shore to account for representativity error.

## 4. Evaluation

The evaluation of RIOPSv2 will be made in three parts. First, an evaluation of the innovations (model-minus-observation differences) will be presented for both RIOPSv2 and GIOPSv3 to demonstrate that the adaptations of SAM2 for the RIOPSv2 system are performing well and to highlight the areas of improvement and degraded performance. We then assess the variations

in sea level anomaly along a particular Jason satellite altimetry track over the 3-year evaluation period to show specific differences in the quality of the two analysis systems. Finally, we compare the power spectral density of the kinetic energy fields to investigate the representation of smaller-scale oceanic features in the RIOPS analyses. We begin by describing the experimental setup used to produce the analysis cycles being evaluated.





## 4.1 Experimental Setup


To support the evaluation, a series of weekly (RD) analysis cycles were produced over a 3-year period (09-Sep-2015 to 02-Jan-2019). The simulations are initialized from rest on 09-Sep-2015 from the World Ocean Atlas Climatology 2013v2 (Boyer et al., 2013) temperature and salinity fields and ice fields are taken from the forced model simulation used to produce the error modes. Following six weeks of spin-up (i.e. 6 cycles), the ocean data assimilation scheme is activated. This approach allows

the online harmonic analysis sufficient time to converge prior to assimilating SLA. The system is then run for 10 cycles to allow the system to stabilize and the amplitude of innovations to reduce as fields are adjusted towards current conditions. The evaluation is then performed on the subsequent cycles run from 06-Jan-2016 to 02-Jan-2019.

Atmospheric forcing is applied in the same manner as used in operations, that is, using the lowest-level atmospheric forecasts

fields from the Regional Deterministic Prediction System (RDPS; 10 km grid resolution) blended with fields from the Global Deterministic Prediction System (GDPS; 25 km grid resolution) to cover the full RIOPS domain (for details see Dupont et al., 2019). Forecast fields from subsequent twice-daily forecasts (i.e. at 00Z and 12Z daily) at lead times of 6-24 h are blended together with a 6-h linear interpolation window to provide a continuous atmospheric forcing set. This approach minimized potential shocks in atmospheric pressure fields due to variations in subsequent forecasts. Note that errors found here (in

particular SST biases) may be larger than found in the real-time operational RIOPSv2 due to improvements implemented in the RDPS and GDPS; in particular those associated with the use of GEMv5 (McTaggart-Cowen et al., 2019) following the implementation on 03-Jul-2019.

The reference set of simulations used here for comparison are from the GIOPSv3 system. This was chosen as a reference since

it uses an equivalent ocean data assimilation system and thus provides an assessment focused on the regional adaptations introduced here for RIOPSv2. Moreover, the RIOPSv1 system constrained temperature and salinity fields towards GIOPS using a spectral nudging approach. Thus, comparison of RIOPSv2 to GIOPSv3 provides a proxy for differences between RIOPSv2 and RIOPSv1. Direct comparison of RIOPSv2 and RIOPSv1 using innovation statistics is not possible since the innovations are calculated online and RIOPSv1 did not use SAM2. The GIOPSv3 simulation used here for comparison was

initialized and spun-up using the exact same methodology as used for RIOPSv2. For both RIOPSv2 and GIOPSv3, the SLA observations used during the evaluation period include Saral/Altika, Jason2 (until Sep 2016), Jason 2N (from Feb. 2017 to Jun. 2017), Jason 3 (from Sep. 2016 onwards) and CryoSat2. With at least three radar altimeters working at any given time, this provides a fairly data-rich period for SLA with which to evaluate the assimilation system.

## 4.2 Innovations

Here we present innovation statistics calculated over the 3-year evaluation period for SLA, SST, and temperature and salinity profile data. As noted in Section 3.1, SAM2 uses an FGAT approach to calculate innovations at the closest model time step.

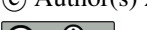



To provide a spatial representation of the innovation statistics, innovations were binned on a 1° latitude-longitude grid. The mean and root-mean-squared (RMS) differences were then produced for each evaluation grid point for innovations covering the full 3-year period. For profile data, innovations were evaluated over different depth ranges. For brevity, here we show results over the depth ranges 0-500 m and 500-2000 m.

### 4.2.1 SLA innovations

The mean and RMS innovations statistics for SLA are shown in Fig. 9. The largest innovations for both RIOPSv2 and GIOPSv3 occur over the Gulf Stream region due to the associated strong mesoscale variability. In GIOPSv3, RMS differences exceed 25 cm from the east coast of North America eastward to 45°W. RIOPSv2 shows notably smaller RMS differences over this region with values closer to 20 cm. These improvements are due in part to the smaller values of MDT representation error used in RIOPSv2, together with the higher model grid resolution.

A second region of large SLA innovations can be found in the Arctic Ocean. In the northern Laptev and Beaufort Seas, RMS differences larger than 20 cm are present. These regions are also characterized by negative mean SLA innovations of greater than 10 cm in GIOPSv3 and RIOPSv2. As such, the differences may be associated with the MDT fields used. As this area is typically ice covered, the MDT would have had few years of open water data to use in its production. These regions are also strongly affected by freshwater drainage from the Mackenzie River and rivers in Russia that may affect the SLA. For example, a negative SLA bias of 10 cm (positive values in Fig. 9 (c) and (d)) at the mouth of the Lena River (135°W) is likely due to positive anomalies in actual river runoff as compared to the climatological values used here by both systems. Note also that due to satellite orbits and ice coverage, many fewer observations are present over the Arctic Ocean, with fewer than 500 measurements per bin as compared to over 2000 measurements per bin south of 66°N. As such, the statistics over the Arctic are less reliable and more seasonally dependent.

Finally, a region with slightly increased RMS differences in RIOPSv2 can be noted in Hudson Bay and the northern Labrador Sea. These regions show larger RMS differences of up to 5 cm and are likely associated with residual tidal variability remaining after application of the online harmonic analysis due to fortnightly and monthly timescale tidal harmonics.

### 4.2.2 SST Innovations

SST innovations between RIOPSv2 and GIOPSv3 are broadly similar (Fig. 10), with the largest errors occurring in the Gulf Stream, and along the winter marginal ice zone in the Labrador Sea and Greenland Iceland Norwegian Seas. RMS SST innovations in the Gulf Stream region decrease from upwards of 2.0°C for GIOPSv3 to around 1.5°C for RIOPSv2. The largest SST innovations for both systems occur along the tail of the Grand Banks around 45°W, likely due to the presence of strong SST gradients. Along the region of maximum ice extent in winter, errors of around 0.8°C are present. In most other areas,





errors in both systems are less than 0.5 °C, the nominal error of the CCMEP SST analysis (Brasnett and Colan, 2016). Similar

to innovations of SLA (Fig. 9), a slight increase in RMS SST innovations in RIOPSv2 can be seen in the central Labrador Sea.

### 4.2.3 T/S profile innovations

Innovation statistics obtained from in situ temperature and salinity profiles averaged over the upper 500 m of the water column and over the range 500-2000 m are shown in Figs. 11-14. These observations are composed mainly of data from Argo profiling floats, with some additional profiles from field campaigns, moorings, voluntary observing ships, gliders and marine mammals.

Interestingly, the period of evaluation includes the Year of Polar Prediction (2017-2019; Jung et al., 2016) for which a number of additional in situ ocean observations were deployed (Smith et al., 2019b). These include Argo and ALAMO floats (Wood et al., 2018) used seasonally during ice-free periods as well as Ice Tethered Profilers (ITP; Toole et al., 2011). These, together with other additional ocean observations deployed during YOPP, provide an exceptional opportunity to evaluate water mass properties in RIOPS and GIOPS over Arctic Ocean, for which a significant gap is usually present in the in situ ocean observing

network.

As shown for SLA and SST, the largest innovations in profile data for both systems are found in the Gulf Stream region due to the strong mesoscale variability and important spatial variability in water mass properties (Figs. 11 and 13). Significant innovations in salinity are also found along many coastal regions. On the Canadian east coast, in the Gulf of St. Lawrence and

along the Labrador Coast, RMS salinity innovations often exceeding 0.5 psu are present. Similar errors are present around the North Sea and along the Norwegian coastline with slightly larger values in RIOPSv2. The larger biases in RIOPSv2 may be associated with a positive salinity bias in the upper 50 m of the water column (Fig. 12). This bias may be due to a deficit in river runoff together with the associated reduction in vertical stratification resulting in overly intense vertical mixing due to tides. The use of a single prognostic equation turbulent kinetic energy scheme may contribute to this effect as well.

Investigations into potential improvements by use of a k-ε turbulent mixing scheme (Umlauf and Burchard, 2003) are underway.

The Arctic observations are quite revealing, highlighting important positive biases in salinity (i.e. too salty) in the Beaufort Sea (Fig. 12c) associated with the halocline (centred at 50 m depth) and Pacific water layer (200-300 m depth; not shown).

This bias is smaller in GIOPS in the eastern Beaufort Sea, with a salinity bias of less than 0.2 psu, whereas RIOPSv2 shows values around 0.4 psu. The source of this bias is under investigation and will be part of a study to investigate how to better constrain water masses in the Arctic using observations from YOPP.

Errors in other regions are generally quite small in both systems. In the Pacific Ocean, the 3DVar bias correction scheme

reduces considerably the biases in salinity over the upper 500 m. Without the bias correction scheme, errors of up to 0.5 psu occur over a broad region in the North Pacific Ocean along the Aleutian Islands (not shown).





### 4.2.4 Evaluation of sea level anomalies over the Gulf Stream region

As the Gulf Stream region shows the largest RMS innovation errors for all fields, we now focus on a comparison of SLA anomalies and resolved spatial scales over this region. Figure 15 shows Hovmöller diagrams for a particular Jason altimeter track illustrating the evolution of SLA anomalies over the evaluation period. The vertical axis is constructed using the SLA anomalies over the track from the 10-day repeat orbit with each row representing one track. The location of the track is shown in green in Fig. 15(b). Both RIOPSv2 and GIOPSv3 capture well the low-frequency variations in SLA due to the evolution of mesoscale ocean features in this highly dynamic region. Indeed, SLA variations are present with a range greater than 1 m, yet

RMS differences for RIOPSv2 and GIOPSv2 are only 12 cm and 14 cm respectively. Differences between either analysis system and the observations (Fig. 15 (d) and (f)) tend to be quite localized in time, likely representing the delay in the analysis system in adjusting to SLA errors. The differences occur less frequently in RIOPS, suggesting a better representation of the mesoscale eddy field. In addition, GIOPS shows a significant period of error related to a negative SLA anomaly that occurred around 62°W in mid-2017. The closer fit of RIOPS to observations is reflected in higher correlations and lower RMS error

between RIOPS and the SLA observations (Table 3).

Given the higher spatial resolution of RIOPSv2, one would expect to see finer scales present in the details of the along-track SLA anomalies. As this is not obvious from Fig. 15 we will now investigate the power spectral density (PSD) of the surface kinetic energy (KE) fields from RIOPSv2 and GIOPSv3 over the Gulf Stream region (Fig. 16) following the approach of

Jacobs et al. (2019). First, we compute the Fourier Transform of both zonal and meridional surface current fields over a box covering the Gulf Stream region (48.8-66.0°W, 32.3-45.0°N) using weekly instantaneous SLA fields from both RIOPSv2 and GIOPSv3 over the full 3-year evaluation period. The results are then averaged in time and then between velocity components to provide the PSD of KE.

Overall, the surface kinetic energy for RIOPSv2 shows 1.7 times the power over the full spectrum with higher power at all wavelengths. The PSD for both systems follows closely the $k^{-3.4}$ line over the mesoscale band in agreement with Jacobs et al. (2019) demonstrating a good representation of the energy cascade. The wavelengths for which the PSD deviates from the $k^{-3.4}$ line at both the high and low wavelength limits is indicated in Fig. 16. Due to the higher model resolution, RIOPSv2 extends the mesoscale band down to about 35 km as compared to 66 km for GIOPS. These scales provide an indication of the effective

model resolution, and are equivalent to roughly 5 and 3 model grid points for RIOPSv2 and GIOPSv3 respectively. The long-wavelength limit for RIOPSv2 and GIOPSv3 are broadly similar with values of 178 km and 208 km respectively. Both systems show a peak in PSD around 416 km.



## 5. Summary and discussion

Here we present a description and evaluation of the first pan-Canadian ocean analysis system running operationally as part of
RIOPSv2. This system includes various improvements with respect to its equivalent global ocean analysis system, GIOPSv3.
These improvements include: a 7-day IAU procedure, the use of higher-resolution background error modes and a spatial filter
used as part of the sea surface temperature observation operator. The most notable change, however, is the inclusion of tides
which require an online harmonic analysis of sea surface height as part of the observation operator. A new method is presented
here that makes use of a sliding-window approach. This approach allows for time-varying harmonic constants that can adapt
to seasonal variations in the tides due to the sea ice cover (e.g. Kleptsova and Pietrzak, 2018).

An evaluation was presented of innovations of SLA, SST and in situ temperature and salinity profile observations for RIOPSv2
and GIOPSv3 over a 3-year period. The results showed similar overall innovation statistics between the two systems,
demonstrating that the use of explicit tides and the online tidal filter did not lead to any significant degradation in the quality
of the analyses. The largest errors for both systems occur in the Gulf Stream region for all fields, with smaller RMS innovations
for RIOPSv2 of about 5 cm for SLA and 0.5 °C for SST.

Some areas did show larger innovation statistics for RIOPSv2 highlighting areas for improvement. In the Arctic Ocean, larger
mean and RMS SLA errors are found for RIOPSv2 close to the North Pole in the marginal ice zone of the seasonal sea ice
minimum. These errors may be related to errors in MDT as significant mean innovations are present and fewer observations
were available to construct the MDT in these regions. These errors in the central Arctic may be expected to occur more
frequently in the real-time operational systems in future years, as declining summer sea ice cover in the Arctic will lead to
increasing areas of open water.

Some increase in SLA innovations (up to 5 cm) locally in areas of Hudson Bay and Baffin Bay are also found. As these regions
experience strong tides with important seasonality and baroclinic effects (Saucier et al., 2004), the increase in SLA innovation
statistics may be due to unfiltered tidal residuals related to fortnightly baroclinic modes. Extending the online tidal filter
approach from a 1D analysis to take into account 2D tidal correlations may be able to improve this somewhat by making it
possible to increase the number of constituents (i.e. to include fortnightly and monthly) without erroneously filtering mesoscale
variability.

As part of YOPP (2017-2019), a significant number of in situ temperature and salinity observations were taken and made
available via the Global Telecommunications System allowing their use in studies such as these that rely on operational
databases. These valuable observations provide a rare glimpse at errors in Arctic water mass properties (Smith et al., 2019b).
Here, we see RIOPSv2 shows larger salinity errors in the Beaufort Sea, with salinity biases of 0.3-0.4 psu in the eastern



Beaufort Sea. Larger salinity biases are also present along many coastlines for RIOPSv2, in particular in the Baltic and North Seas. These errors highlight the need for a more sophisticated turbulent closure model needed for use with explicit tides. The addition of a k-ε closure model (Umlauf and Burchard, 2003) is under investigation.

As the largest innovations in both RIOPSv2 and GIOPSv3 occur in the Gulf Stream region, a comparison of SLA along a particular Jason satellite track was made. Both systems capture the main evolution of mesoscale variations, however, RIOPSv2 shows smaller differences from SLA observations than GIOPSv3 (Table 3) with higher correlation (0.82 and 0.73 respectively) and lower RMS (12 cm and 14 cm respectively).

To investigate the role of increased resolution in RIOPSv2, the PSD of surface kinetic energy over the Gulf Stream region was investigated. The PSD of RIOPSv2 is 1.7 times the power of GIOPSv3 and contains smaller-scales down to roughly 35 km as compared to 66 km for GIOPSv3. These scales give an indication of the effective model resolution and correspond roughly to 5 and 3 grid points respectively. These values are somewhat smaller than one would expect purely from a numerical point of view (generally the smallest feature resolvable requires 6-10 grid points) and suggests the data assimilation may be artificially
increasing the resolution somewhat. This is not surprising for GIOPSv3, as it has only a 1/4° grid resolution and is thus considered eddy-permitting only.

The evaluation presented here demonstrates that RIOPSv2 is able to produce analyses of a similar quality to GIOPSv3 albeit including higher grid-resolution and explicit tides. However, the evaluation is restricted to the observations assimilated by the
systems and provide only indirect information regarding the quality of related quantities such as surface currents, which are important for key applications of RIOPSv2 such as emergency response. An alternative means to evaluate the skill of surface currents is to use drifting buoys. A significant effort is underway as part of the OPP initiative to deploy various types of drifters in Canadian waters to evaluate numerical ocean predictions. As part of this effort, a comparison of RIOPSv1.3 and TOPAZ with drifter observations off the coast of Norway showed positive results (Sutherland et al., 2020).


Despite being an important application for RIOPSv2, quantitative evaluation for emergency response is especially challenging due to a lack of cases with sufficient data. Qualitative evaluation from respondents from the emergency response section at CCMEP have found that RIOPSv2 provides significant improvements in many areas. However, in cases close to coast, RIOPSv2 applicability is limited due to the relatively coarse ~3-6 km grid along the Canadian coastline. As a result, two sub-
domains at 2 km grid-resolution that downscale RIOPSv2 fields have been developed and are being run experimentally at CCMEP to evaluate the added value of increased resolution. An evaluation of RIOPSv2 analyses for the Canadian east and west coasts is being made as part of this effort that includes comparison with drifting buoys, acoustic Doppler current profiler observations and high-frequency radar observations.





Another important application of RIOPSv2 is for sea ice prediction. An evaluation of forecasts of sea ice concentration, drift and thickness showed very little change from RIOPSv1.3 to RIOPSv2 (Dupont et al., 2019). The only significant impact is a slight reduction in the negative bias in ice drift due to the reduction in the ice-ocean drag coefficient. The lack of impact on sea ice is not surprising given the similarity between RIOPSv2 and GIOPSv3 analyses and that RIOPSv1.3 used a spectral nudging approach toward GIOPS analyses. As such, differences in ocean fields (e.g. SST, mixed-layer depth) in the vicinity
of the sea ice were quite small and thus had little impact of the evolution of sea ice fields (e.g. due to formation, melt).

**Code and data availability**

The codes, scripts and data used in this paper are available for the topical editor and anonymous reviewers.

**Author contributions**

GS, YL, MB, CT, FDu, JLe, FDa were responsible for the concept. GS contributed to writing and original draft preparation.
GS, CT, FDu, JLe contributed to writing, review and editing. FDu and JLi developed the extended model domain. YL, FDu, MB, CT and FR were responsible for software. YL and KC performed the numerical experiments and system evaluations.

**Competing interests**

The authors declare that they have no conflict of interest.

**Acknowledgements**

The authors acknowledge the support of Pierre Pellerin, Hal Ritchie, Youyu Lu and other members of the CONCEPTS consortium. They also thank Jean-Philippe Paquin for the suggestion to examine SLA Hovmöller diagrams. They also acknowledge the support of the informatics and operational implementation groups at CCMEP. In situ temperature and salinity profile data from Argo floats used here were collected and made freely available by the International Argo Program and the
national programs that contribute to it. (http://www.argo.ucsd.edu, http://argo.jcommops.org). The Argo Program is part of the Global Ocean Observing System. This study has been conducted using E.U. Copernicus Marine Service Information. This study is a contribution to the Year of Polar Prediction (YOPP), a flagship activity of the Polar Prediction Project (PPP), initiated by the World Weather Research Programme (WWRP) of the World Meteorological Organisation (WMO). We acknowledge the WMO WWRP for its role in coordinating this international research activity.



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





## Appendix A

**Derivation of real-space equations for the sliding-window online harmonic analysis**

In RIOPSv2, we work on the real vector space spanned by the real function base pair of $\cos(\omega_k \tau n)$ and $\sin(\omega_k \tau n)$. To replace
the complex space spanned by $e^{i\omega_k \tau n}$ as described in Section 3.4.2 we employ the Euler identity $e^{i\theta} = \cos\theta + i\sin\theta$. Its dimension becomes *2K+1*, including the 1-dimensional real subspace for $\omega_0 = 0$. The complex space matrices **R** and **B** are replaced by real space matrix **S** and **D** with size *(2K+1) x (2K+1)*, respectively. Similarly, complex space vector **Y** is replaced by real space vector **Z** with size *(2K+1) x 1*. This is just a transform of the operators' representative space from complex space to the equivalent 2-dimensional (2D) real space.


Because **R** is a diagonal matrix in the *K+1*-dimension complex space, there is no frequency mixing when operating by **R**. This means when working in the *2K+1* dimensional real vector space, the 2D real space, which is spanned by each real function base pair, that this is a complete invariant subspace when operating under **S**. In addition, when $\omega_0 = 0$, it is a 1D complete invariant subspace. In other words, the *2K+1* dimensional space that **S** operates on is reducible, and it can be reduced into a
direct-sum space of one 1D-invariant-subspace and *K* 2D-invariant-subspaces. In doing so, **S** would be reduced and represented as a block-diagonal matrix as following

$$S = S^0 \oplus \sum_{k=1}^{K} \oplus\, S^k \ . \tag{A1}$$

Here, $\oplus$ is the matrix direct-sum operator, $\boldsymbol{S^0}$ is the real number 1 for $\omega_0 = 0$ and $\boldsymbol{S^k}$ is a 2D real rotation matrix with rotation angle $\omega_k \tau$. Therefore, we can rewrite Eqs. (9), (11b) and (6b) as the following *2K+1*-dimensional real space vector and matrix
format

$$Z = \alpha U H^0 + (1-\alpha) S^T Z' \ , \tag{A2}$$

$$D = \alpha 1_c \ + (1-\alpha) S^T D' S \ , \tag{A3}$$

$$A^0 = \sum_{cos}[D^{-1} Z]. \tag{A4}$$

Here, **U** and $\boldsymbol{I_c}$ are a real vector and matrix with element equal to 0 if the element involving sine dimension, otherwise, element
equal to 1, because both $1_{kj}$ and $V_k$ are real number 1 in Eqs. (9) and (11b); $S^T$ is the transpose of **S**; and symbol $\Sigma_{cos}$ denotes the summing over only the cosine components of *2K+1*-dimensional vector $\boldsymbol{D^{-1}Z}$.




**FIGURES AND TABLES**


**Table 1: Changes to Model parameters between RIOPSv1.3 and RIOPSv2.0**

| Parameter | RIOPSv1.3 | RIOPSv2.0 |
|---|---|---|
| Domain | Arctic/N. Atlantic | N.Pac/Arctic/N. Atlantic |
| NEMO Version | NEMOv3.1 with various code modifications back-integrated from NEMOv3.6 | NEMOv3.6 |
| Restarts | RPN standard format (in-house format) | NetCDF |
| I/O format and method | DIMG (sequential) | Netcdf (XIOS in parallel) |
| River temperature | Spread horizontally over selected points | Closest model point |
| River application | Top model level | Spread over vertical |
| St. Lawrence River | Freshwater only | 1D river model |
| Atm-Ice roughness | 0.16 mm | 0.2 mm |
| Ice/ocean roughness | 2.6 cm | 2 cm |
| Background vertical eddy viscosity | $10^{-4}$ m$^2$ s$^{-1}$ | $10^{-6}$ m$^2$ s$^{-1}$ |
| Background vertical eddy diffusivity | $10^{-5}$ m$^2$ s$^{-1}$ | $10^{-7}$ m$^2$ s$^{-1}$ |
| Vertical levels | 50 | 75 |





**Table 2: Model parameters used in RIOPSv2 and GIOPSv3**

| Parameters | RIOPSv2 | GIOPSv3 |
|---|---|---|
| Model | The ocean engine from NEMO3.6 is OPA (Océan Parallelisé; Madec et al., 1998; Madec and NEMO team, 2008; https://www.nemo-ocean.eu)<br>The sea ice component is CICE 4.0 (Hunke, 2001; Lipscomb et al., 2007; Hunke and Lipscomb, 2010). | Same as RIOPSv2 |
| Domain | Regional, from 25.6°N in the North Atlantic Ocean to 43.8°N in the Pacific Ocean. Covers the three Canadian oceans: Part of North Atlantic, Arctic and part of North Pacific Oceans (Fig. 1). | Global |
| Horizontal resolution | 1/12° nominally (from 8 km in the North Atlantic to 3 km in the Canadian Arctic Archipelago (Fig. 1).<br><br>1580 x 2198 horizontal grid points. | 1/4° on global tri-polar ORCA025 grid.<br><br>1441 x 1021 horizontal grid points. |
| Vertical sampling | 75 z-levels | 50 z-levels |
| Numerical technique | Primitive equations with finite differences: Arakawa C-grid in the horizontal and vertical directions. | Same as RIOPSv2 |
| Time integration | Explicit leapfrog, Non-linear free surface solved explicitly (time-splitting approach of barotropic and baroclinic time-stepping).<br><br>Baroclinic time-step : 300 s | Same as RIOPSv2 apart from baroclinic time-step of 450 s. |
| Prognostic variables | Three-dimensional horizontal currents, potential temperature, salinity and turbulent kinetic energy (TKE). 2D field of sea surface height (SSH). | Same as RIOPSv2 |
| Geophysical variables | Bathymetry from ETOPO2 plus some smoothing to accommodate high tides. | Bathymetry derived from ETOPO2. |
| Horizontal diffusion (explicit) | Bi-Laplacian (Del-4) on momentum variables along geophysical coordinate and Laplacian (Del-2) applied on tracers along iso-neutral surfaces. | Same as RIOPSv2 |
| Surface scheme | Bottom atmospheric model level used for flux calculations. CORE bulk | Same as RIOPSv2 |



| | | |
|---|---|---|
| | formulae (Large and Yeager, 2004) for turbulent sensible and latent heat, and momentum. Stability functions adjusted to use time-varying height of bottom atmospheric model level. | |
| **Turbulent mixing (vertical diffusion).** | "TKE" scheme based on Gaspar et al. (1990) and Blanke and Delecluse, (1993). | |

**Table 3: SLA statistics for GIOPS and RIOPS as compared to a particular Jason satellite altimetry track across the Gulf Stream (shown in Fig. 15 (b))**

| | GIOPSv3 | RIOPSv2 |
|---|---|---|
| **Correlation** | 0.73 | 0.82 |
| **Std. Dev. (cm)** | 13.93 | 11.87 |
| **RMS (cm)** | 13.94 | 11.97 |




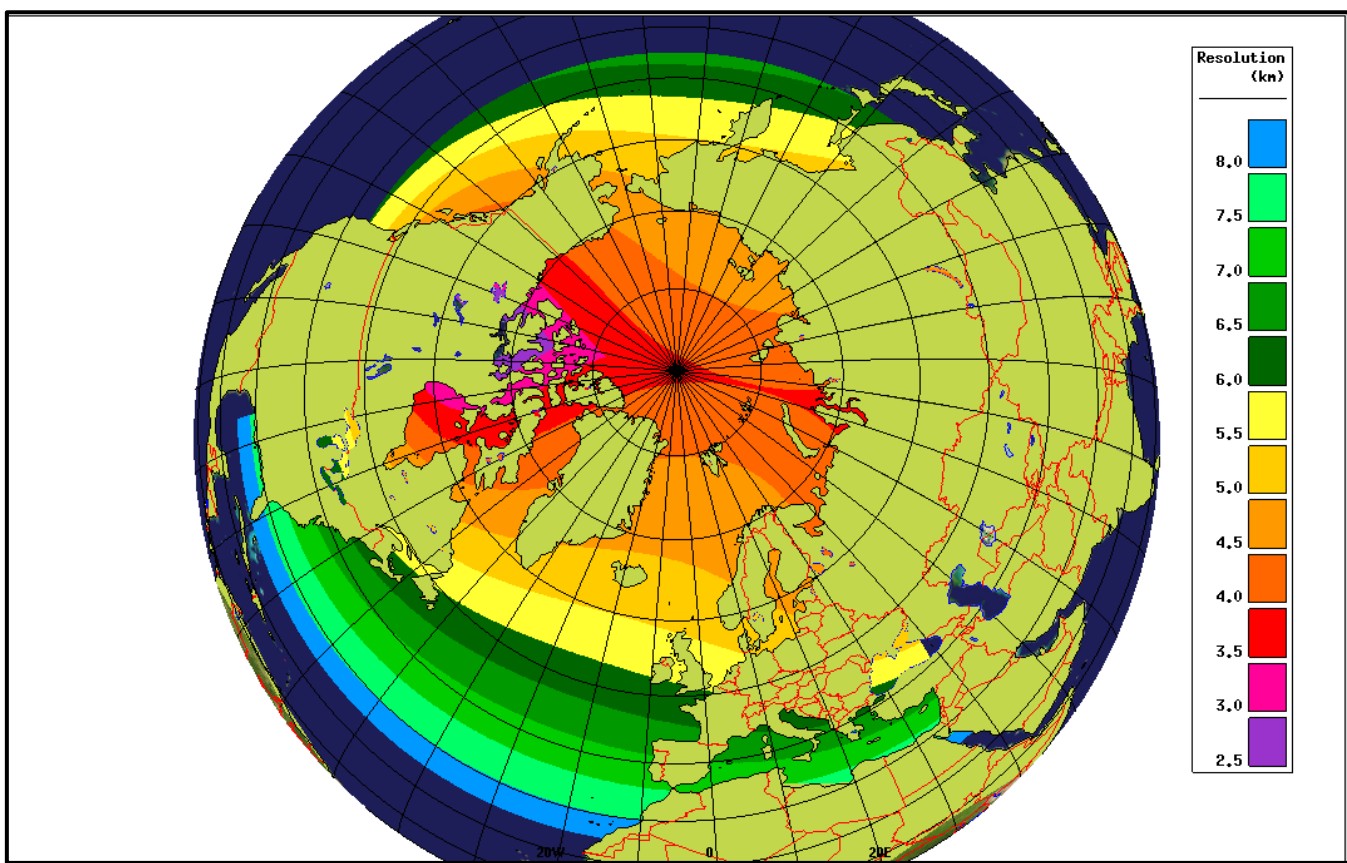

*Figure 1: The coverage of the CREG12 extended domain used in RIOPSv2 showing the model grid resolution (km). The domain*
*extends from 26°N in the Atlantic Ocean over the Arctic Ocean to 44°N in the Pacific Ocean. The model bathymetry is set to*
*zero for the partially-covered regions of the Gulf of Mexico, the Black and Red seas as well as for inland lakes.*





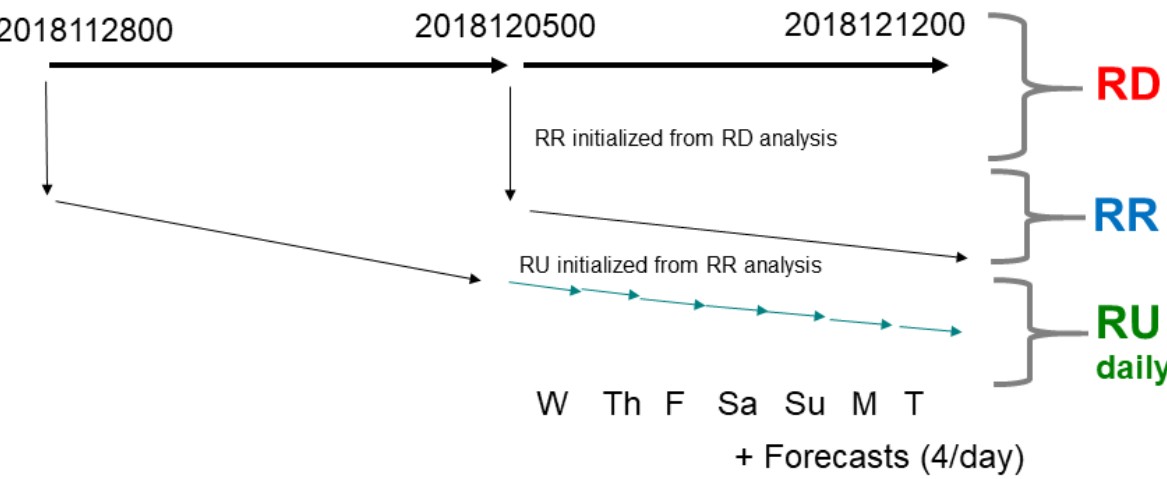

*Figure 2: Schematic diagram showing the sequencing of the delayed-mode (RD), real-time (RR) and daily update cycles (RU). The RD cycle is run 7 days behind real-time each Wednesday and provides continuity in time. The RR cycle is initialized from the RD cycle and provides a real-time analysis each Wednesday. Finally, the RU cycles provide daily updates using a shorter 1-day analysis cycle assimilating only sea ice and sea surface temperature. Additionally, 48 hr forecasts are produced from RU analyses 4 times per day (00Z, 06Z, 12Z, 18Z).*



*Figure 3: Example of spatial structures of multi-variate correlations from background error modes for Sept. 15 in the Gulf Stream region. The left panels show the spatial correlation of SST with the point marked with a star over the full domain (top left) and over a magnified region (bottom left). The right panels show the spatial correlation of SST with 3-D temperature (top right) and the spatial correlation of SSH with 3-D temperature (bottom right). The 3-D cube is plotted such that the bottom left corner corresponds to the point marked with the star. The vertical dimension is plotted using the model level to enhance the resolution near-surface. Levels 20 and 40 correspond roughly to 60 m and 200 m depth respectively. The spatial extent of the cube is shown in the left panel as a magenta box. The bubble radius used in the localization procedure to reduce long-range spurious covariances is shown as a dashed oval. Model bathymetry is shown in grey. An animated version of this figure is available for which the central point marked with a star moves along the magenta line (supplementary material).*



Figure 4: *Same as Fig. 3 for a point off the Labrador Coast.*


Figure 5: *Same as Fig.3 for an off-shore point in the Beaufort Sea.*



*Figure 6: Same as Fig. 3 for an on-shore point in the Beaufort Sea.*



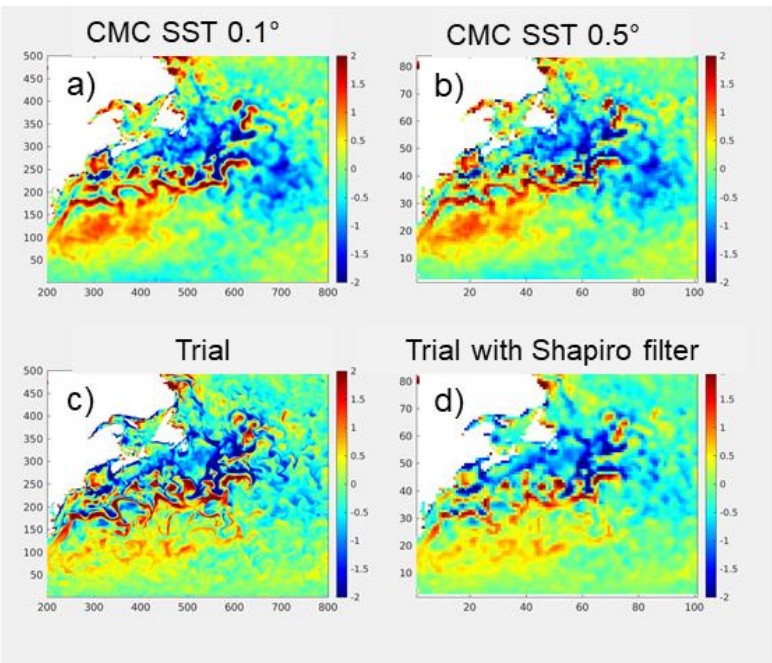

*Figure 7: Example illustrating the SST filtering made as part of the observation operator in RIOPSv2 for 20-Jul-2016. The*
*CCMEP SST analysis assimilated by RIOPSv2 is shown on its native 0.1° latitude-longitude grid in panel (a). This analysis is*
*first decimated to one-point out of 5 to reduce correlated errors (panel b). The RIOPSv2 7-day trial field is shown in panel*
*(c). Finally, the trial field following application of a Shapiro filter is shown in panel (d). The innovation is calculated as panel*
*(b) minus (d).*



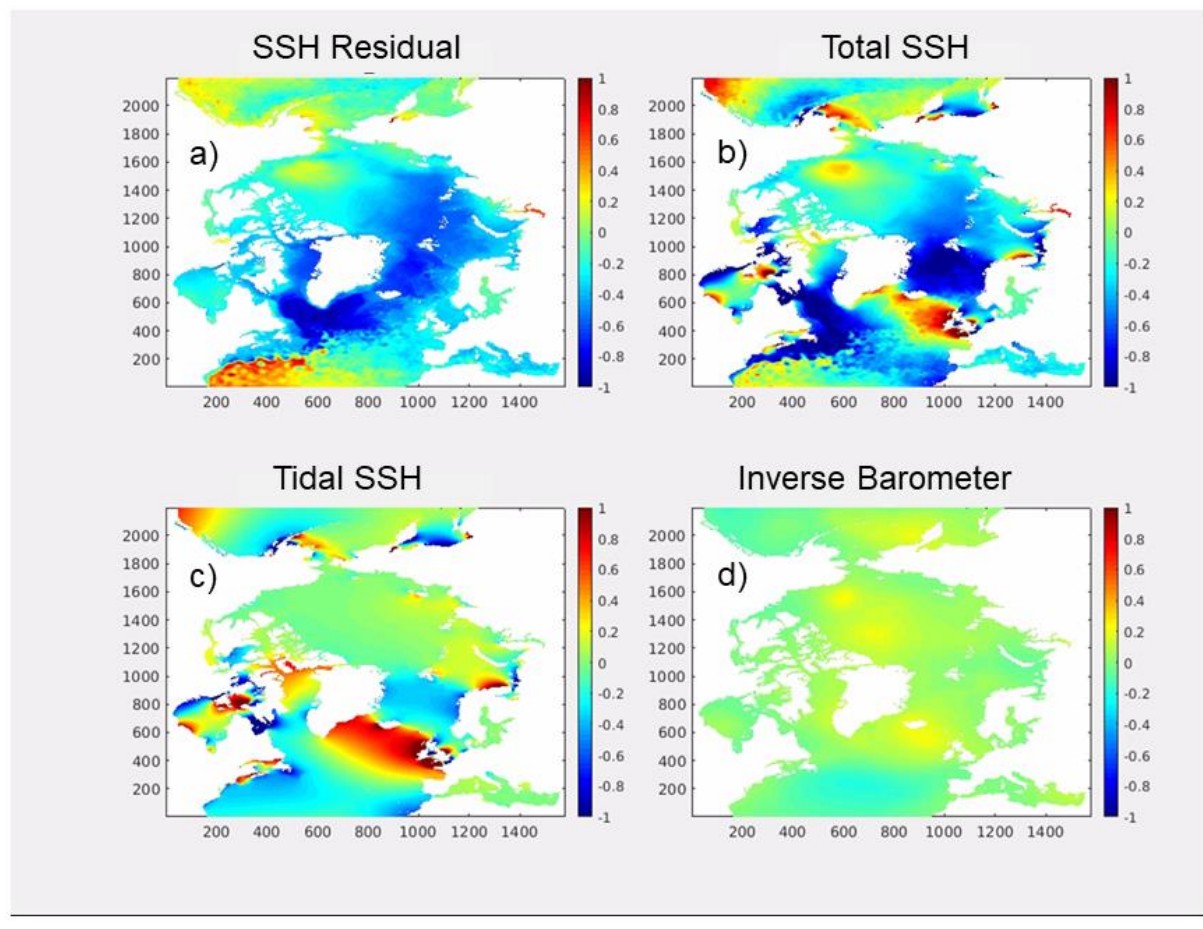


*Figure 8: Example showing impact of tidal and inverse barometer terms on SLA observation operator. The SSH residual used in the SLA observation operator is shown in panel (a). Panel (b) shows the instantaneous model SSH field prior to any treatment. The tidal component calculated using the online harmonic analysis is shown in panel (c). Panel (d) shows the inverse barometer component. Units in m. Fields are plotted on native model grid with grid-point numbers shown on the x and*

*y axes.*





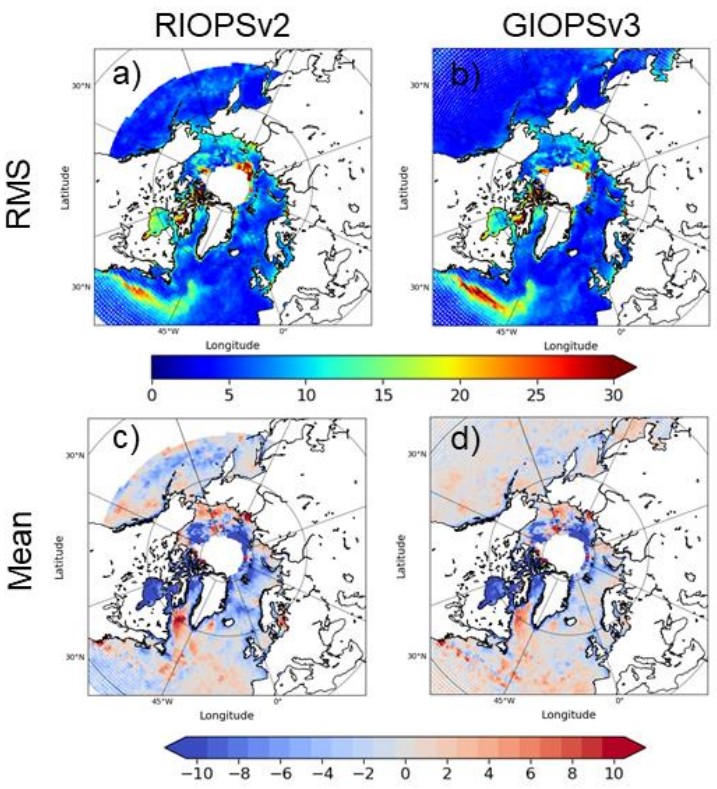

*Figure 9: Innovation (observation-model) statistics of sea level anomaly for the period 01-Jan-2016 to 31-Dec-2018. The RMS (top row) and mean differences (bottom row) are shown for RIOPSv2 (left column) and GIOPSv3 (right column).*


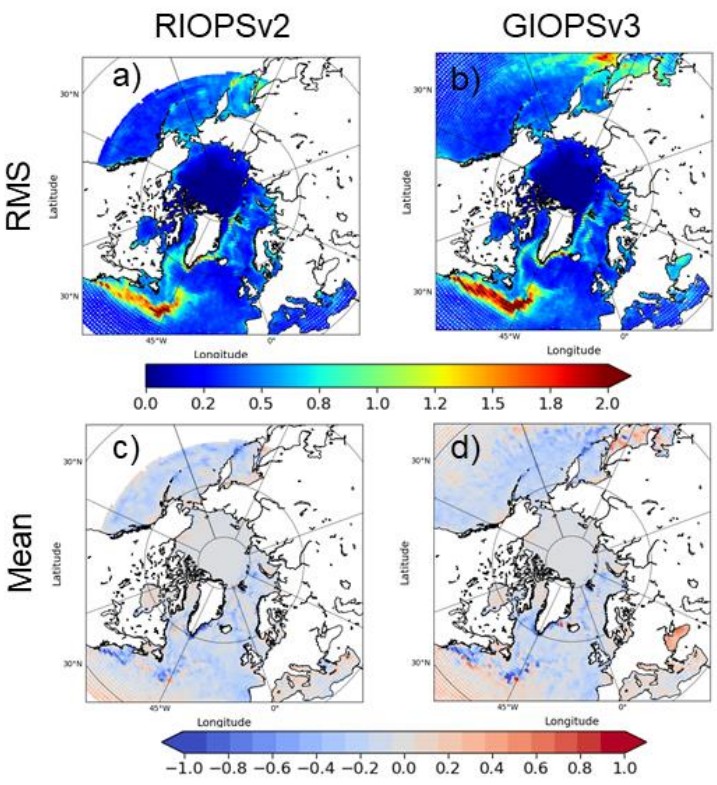

*Figure 10: Innovation (observation-model) statistics of sea surface temperature for the period 01-Jan-2016 to 31-Dec-2018. The RMS (top row) and mean differences (bottom row) are shown for RIOPSv2 (left column) and GIOPSv3 (right column).*



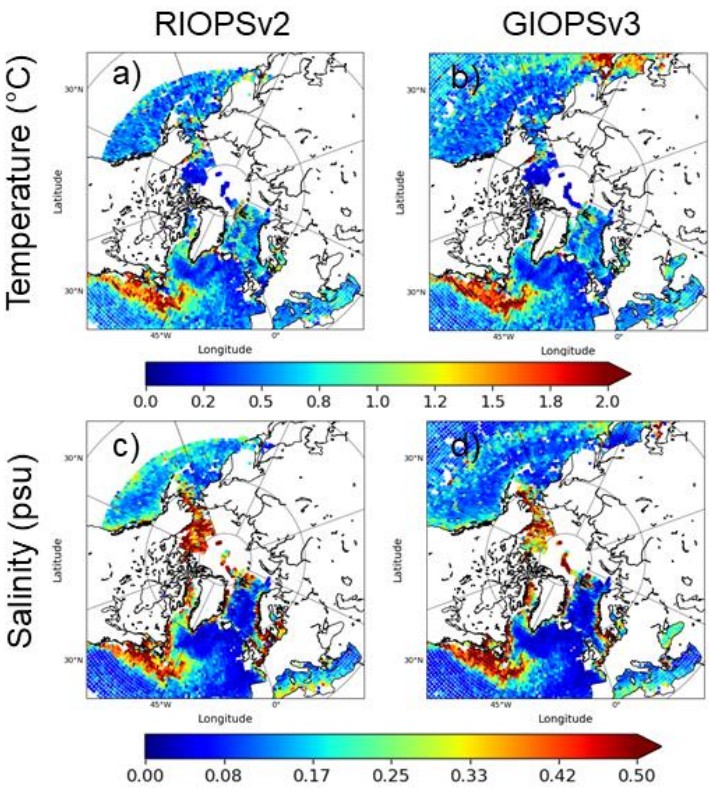


*Figure 11: Innovation (observation-model) statistics of in situ temperature and salinity over the upper 500 m depths for the period 01-Jan-2016 to 31-Dec-2018. The RMS differences for temperature (top row) and salinity (bottom row) are shown for RIOPSv2 (left column) and GIOPSv3 (right column).*



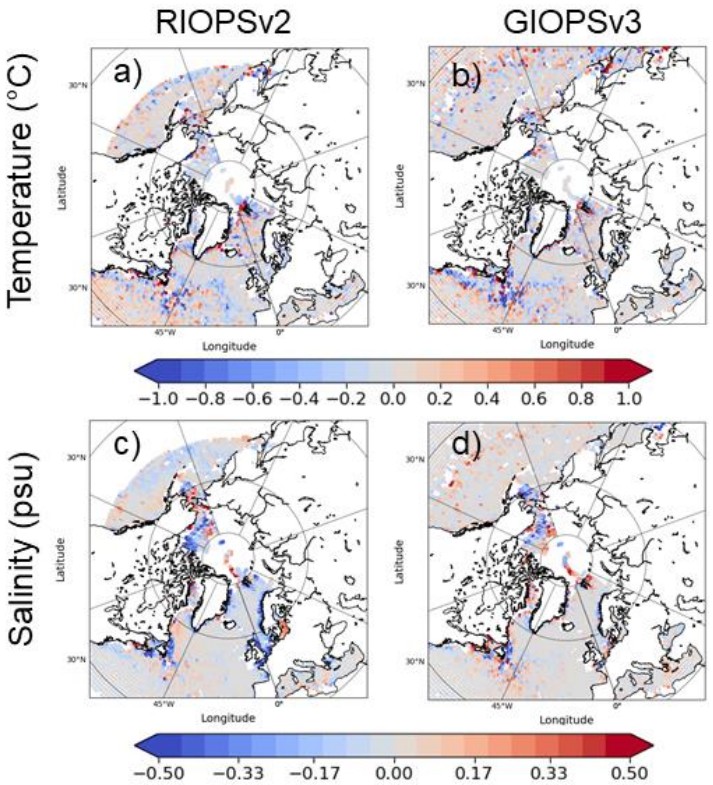


*Figure 12: Innovation (observation-model) statistics of in situ temperature and salinity over the upper 500 m depths for the period 01-Jan-2016 to 31-Dec-2018. The mean differences for temperature (top row) and salinity (bottom row) are shown for RIOPSv2 (left column) and GIOPSv3 (right column).*



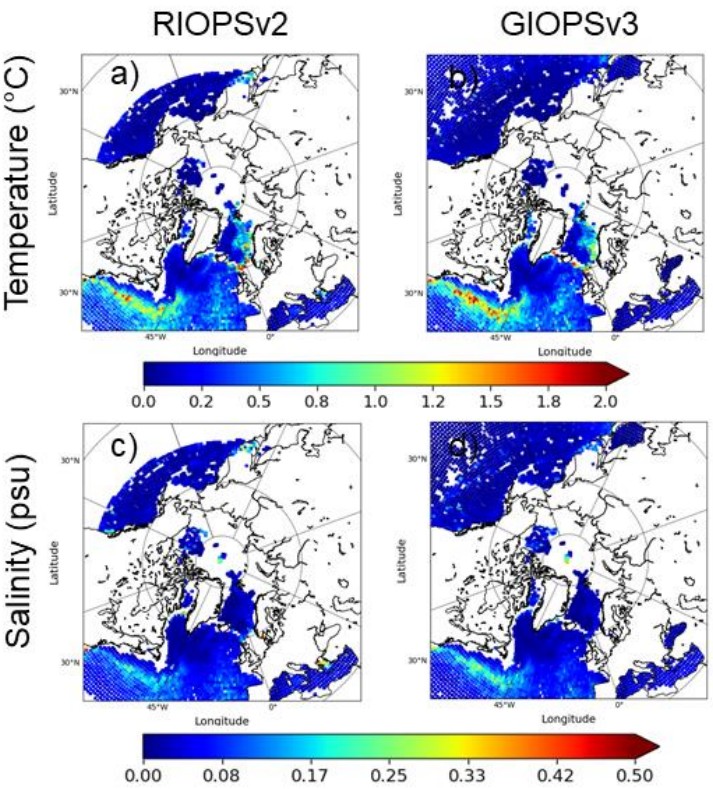

*Figure 13: Innovation (observation-model) statistics of in situ temperature and salinity over the depth range 500-2000 m for the period 01-Jan-2016 to 31-Dec-2018. The RMS differences for temperature (top row) and salinity (bottom row) are shown for RIOPSv2 (left column) and GIOPSv3 (right column).*






*Figure 14: Innovation (observation-model) statistics of in situ temperature and salinity over the depth range 500-2000 m for the period 01-Jan-2016 to 31-Dec-2018. The mean differences for temperature (top row) and salinity (bottom row) are shown for RIOPSv2 (left column) and GIOPSv3 (right column).*



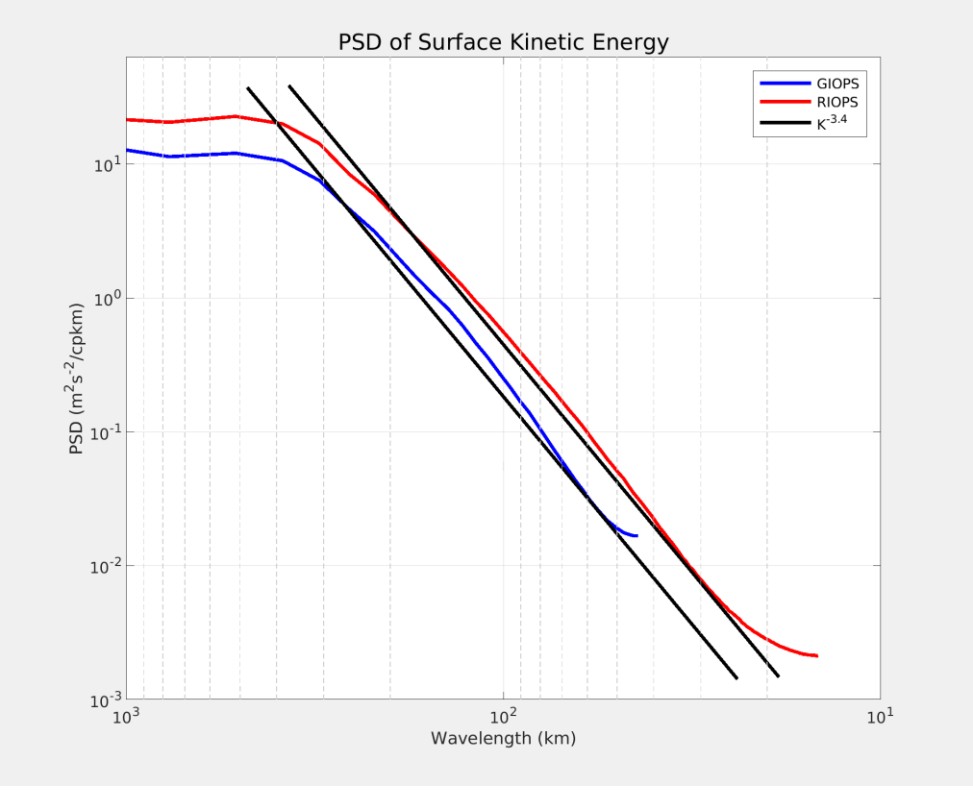


*Figure 15: Hovmöller diagrams showing variations in sea level anomaly for the period 01-Jan-2016 to 31-Dec-2018 along a repeat altimeter track of the Jason altimeter (green line in panel (b)). Observations prior to 07-Sep-2016 are taken from Jason2 and Jason3 is used thereafter. Satellite observed values are shown in panel (a) along with values for RIOPS (b) and GIOPS (c). Differences (obs – model) for RIOPS and GIOPS are shown in panels (d) and (f) respectively. The RMS (top row) and Mean differences (bottom row) are shown for RIOPSv2 (left column) and GIOPSv3 (right column). Periods of missing observations are shown as grey boxes.*
