# Peer review of "The Regional Ice Ocean Prediction System v2: a pan-Canadian ocean analysis system using an online tidal harmonic analysis"

_Geoscientific Model Development, 2020_

## Short Comment (SC1) · 10 Sep 2020

Dear authors,

in my role as Executive editor of GMD, I would like to bring to your attention our Editorial version 1.2:

https://www.geosci-model-dev.net/12/2215/2019/

This highlights some requirements of papers published in GMD, which is also available on the GMD website in the 'Manuscript Types' section:

http://www.geoscientific-model-development.net/submission/manuscript_types.html

In particular, please note that obviously you completely misunderstood out code avail-

ability requirements, as the following point has not been met in the Discussion Paper:

- "Code must be published on a persistent public archive with a unique identifier for the exact model version described in the paper or uploaded to the supplement, unless this is impossible for reasons beyond the control of authors. All papers must include a section, at the end of the paper, entitled "Code availability". Here, either instructions for obtaining the code, or the reasons why the code is not available should be clearly stated. It is preferred for the code to be uploaded as a supplement or to be made available at a data repository with an associated DOI (digital object identifier) for the exact model version described in the paper. Alternatively, for established models, there may be an existing means of accessing the code through a particular system. In this case, there must exist a means of permanently accessing the precise model version described in the paper. In some cases, authors may prefer to put models on their own website, or to act as a point of contact for obtaining the code. Given the impermanence of websites and email addresses, this is not encouraged, and authors should consider improving the availability with a more permanent arrangement. Making code available through personal websites or via email contact to the authors is not sufficient. After the paper is accepted the model archive should be updated to include a link to the GMD paper."

Thus with your statement that the code is available to the editor and referees you do not even meet the minimum requirements. Please make you code publicly and permanently available. If this is not possible due to license reasons, state this explicitly (Who is proprietor of the Code? Which license does prevent publication?) in the "Code Availability" section. Note that this is mandatory to finally publish the article!

Yours,

Astrid Kerkweg

---

## Author Comment (AC1) · 11 Sep 2020

Dear Dr. Kerkweg,

Thank you for your interest in our paper. We acknowledge the code availability requirements of GMD and we would like to applaud GMD on your efforts to make code more openly available to the scientific community. Our intention is adhere to the GMD code availability requirements to the extent possible given our licensing arrangements. I would also like to bring to your attention a discussion on this matter that was held with the Topical Editor (see previous correspondence).

The topical editor proposed that we change the text in our code availability section to be the following: "The ocean data assimilation code (SAM2) was obtained under license

from Mercator Océan International and cannot be distributed publicly. For this reason, the codes, scripts and data used in this paper were grouped in a dataset on Zenodo and made available for the topical editor and anonymous reviewers". I hope you find this solution acceptable.

Please see below an except from our discussion with the topical editor.

Kind Regards, Greg Smith

Explanation provided to the topical editor: The ocean data assimilation code (SAM2) was obtained under license from Mercator Océan International and we are not able to distribute it publicly. For this reason, we indicated in the manuscript that "The codes, scripts and data used in this paper are available for the topical editor and anonymous reviewers". We understand GMD has a strict data availability policy and our intention is to follow what was done for a paper we published in GMD last year using SAM2 (Skachko et al., 2019).

As a result, we have created a dataset on Zenodo (Smith, 2020) with restricted access, available for you and the paper reviewers only. This dataset contains all the model and data assimilation code for both systems described in the manuscript (so-called RIOPS and GIOPS). Also included are the data used in the paper to produce the figures showing innovation statistics (observation-minus-model differences). Most figures are simply maps of mean and root-mean-squared values of these data. However, Fig. 15 examines the power spectral density of surface kinetic energy. The Matlab code and data files to produce this figure are also included. References Skachko, S., Buehner, M., Laroche, S., Lapalme, E., Smith, G., Roy, F., Surcel-Colan, D., Bélanger, J.M. and Garand, L., 2019. Weakly coupled atmosphere–ocean data assimilation in the Canadian global prediction system (v1). Geoscientific Model Development, 12(12), pp.5097-5112. Smith, G. (2020). The Regional Ice Ocean Prediction System v2 [Data set]. Zenodo. http://doi.org/10.5281/zenodo.3978269

---

## Referee Comment (RC1) · Anonymous Referee #1 · 6 Oct 2020

This paper describes and evaluates the high resolution CCMEP analysis which spans the coastline of Canada and the Arctic. The model and assimilation methodology are introduced in relation to the previous regional system. The performance of the system is assessed by an evaluation of background departure statistics against sea level anomaly and in-situ observations, and compared against the Global Ice Ocean Prediction System.

Firstly let me apologize to the authors for the delay to my review and thank the editor for allowing me sufficient time to complete it. I am glad to be able to have thoroughly read the paper and I feel it very well describes the operational system. Overall I am sure this paper is perfectly suited to GMD and should be published subject to some minor corrections/additions.

[Figure]

I will first give my more general comments, and subsequently give the itemized comments I had whilst reading through the paper.

General comments:

The paper is very well written and was enjoyable to read. For a description of an operational analysis system the authors have the balance just about right in terms of the detailed description within this paper and referencing previous work. In some areas I was not clear whether those references would suffice or if they were differences between the previous system and RIOPSv2. For example the boundary conditions used for the regional system are not well highlighted (I assume they come from GIOPS but this isn't explicitly stated?). Also when the change to model levels was mentioned it highlighted that the vertical domain was not well described and may lead to confusion so a sentence or two would go a long way to help the reader here.

The section on the online harmonic analysis was far more in depth than the others. I am not familiar with such methods and therefore not sure where the methodology from the literature finishes and the new science (I think related to the sliding window) starts. If this could be highlighted it would be a useful addition.

More generally what is not discussed is the mismatch between a model that contains tides (needed for the coastal applications) and SLA observations which do not contain tidal signals. However the raw altimetry data would necessarily contain tidal signals, so it would benefit the reader to have a short section on SLA observations and why you aren't using lower level SLA observations.

I was hoping to see more about sea ice performance but the final paragraph is sufficient to let the reader know why it is not shown. I wonder if this could be expanded upon to hint at how a future system might look to improve sea ice predictions?

Finally I would ask that the authors look at the colour scales they are using in their plots. I think a lot of the structures we see are artifacts of the jet colour scheme and

the use of perceptually uniform colour scales would remove such artifacts and make the paper more accessible to colour blind readers.

Detailed comments:

Line 126: I am unfamiliar with these tidal constituents - is this detail necessary and if so should such parameters be described?

Line 137: Please comment on the vertical domain - "deep layers (from 500m" suggests these are the deepest layers in your domain, but I would expect the domain to go down to ∼4000m in the arctic.

Line 170: This has strong similarities to the ECMWF workflow shown in Browne et al. 2019 Figure 3. ::: Browne, P. A., de Rosnay, P., Zuo, H., Bennett, A., & Dawson, A. (2019). Weakly Coupled Ocean-Atmosphere Data Assimilation in the ECMWF NWP System. Remote Sensing, 11(234), 1–24. https://doi.org/10.3390/rs11030234

Lines 209-211: "Another modification required for coupled forecasts was to use 24-h averaged short and long-wave radiation fields to force NEMO-CICE during the analysis cycles such that there is very little diurnal warming present in the ocean analysis". Please can you elaborate on this. Is the system designed to have a strongly damped diurnal cycle in the analysis, or are there reasons for which this was found to be necessary?

Line 215: Smith et al (2015) should be 2016? Or a different paper?

Line 228: Can you clarify - does this mean you apply the increment from the current cycle to the next cycle (in the first 24 hours)?

Figures 3-6: Subplot titles are missing commas, i.e. <SST(r)SST(r+\Delta r)> should be <SST(r),SST(r+\Delta r)>

Figure 3 - star is not visible, please can you add it on top or produce the image at a higher resolution? In fact maybe just change the text to make it clear the point you are

considering is the bottom left corner of the magenta box?

Figure 3 - this is the first time you mention localisation. Please can you elaborate on why you choose the region that you do? Is the localisation a tensor product of horizontal and vertical localisation functions, or is there no localisation in the vertical? Is it a hard cut off or is it smoothly reduced to zero at the boundary of the "bubble" by a Gaussian or Gaspari-Cohn function? (I suspect you will have a reference for this which would be good, but also nice to know if you have had to make any changes here for RIOPSv2).

Figures 3-11,13,14: If possible please replot using a diverging, perceptually uniform colour scale. For instance in Figure 3, $<SST(r)T(r+\Delta r)>$, the current choice of colourscale appears to show a clear change in correlation around layer 10 whereas I think this is an artifact of the rainbow colour scale. Thyng, Kristen M., et al. "True colors of oceanography: Guidelines for effective and accurate colormap selection." Oceanography 29.3 (2016): 9-13. https://doi.org/10.5670/oceanog.2016.66 Hawkins, Ed. "Scrap rainbow colour scales." Nature 519.7543 (2015): 291-291. https://doi.org/10.1038/519291d

Line 278: "model trial field" is not well defined. I suggest replacing with something like "raw model equivalent field"

Figure 7: Please relabel "CMC" -> "CCMEP" and "Trial" -> "RIOPSv2"

Line 299: ", the use of a rotation" -> ", and the use of a rotation"

Line 306: Are you using Einstein notation here so there is an implicit summation over the repeated index k? This should be noted. Oh I now see it introduced on line 320 - is it used before or only from line 320?

Line 311 - the definition of $\omega_k$ is unclear. please look carefully at the wording you use here.

Line 315: Is this a minimisation over only $\hat{X}^k$ ?

Line 338: Prime symbol is not consistent - MS Word issue?

Line 350: "According to the definition of the sliding window weight". Where is this defined? Eqn 3a? Also have you defined explicitly what "its property of normalisation" is?

Figure 8: Can you clarify, possibly in the caption, that (a) = (b) - (c) - (d)? Again, diverging colour scales would be much better here. In fact the total SSH field (b) is confusing: "Panel (b) shows the instantaneous model SSH field prior to any treatment." but its range is (-1,1). Has this already had the MDT removed? I personally am more familiar with considering the MDT for SLA assimilation when using processed SLA observations that have tidal signals removed, so I would like to see precisely where the MDT is required in your methodology.

Line 477/478. Regarding the Gulf stream errors, would you expect the SEEK filter methodology to be able to effectively constrain a region which such variability given the error covariances are coming solely from a climatology and therefore may be far too smooth there? It is nice to see the errors reduced in RIOPSv2.

Line 485: The difference in errors in the Laptev sea is intriging. Could you produce a comparison plot of the two different MDTs used?

Line 488: We should see errors here in salinity too if that is the case? [Refer to my later comment on Figures 11-13]

Line 489-491: Surely this is entirely due to ice cover - polar orbiting satellites should have much richer coverage in the arctic due to their much more regular return period (if you exclude the polar data gap).

Section 4.2.2/Figure 10. Given the broadly similar structures and different scales between Gulf stream errors and other regions it would be nice to see a difference plot and/or a normalised difference plot of 10(a) and (b). This might be too difficult to produce in a reasonable amount of time/effort, but may prove enlightening.

Figures 11-13: They appear to be missing the polar sector from 45E to 180E. Is this a plotting error? Are there no in-situ obs in the Hudson bay? I was hoping to see salinity departures which would corroborate interpretation of errors around the mouth of the Lena river that you associated to the fresh water fluxes.

figure 11 - Salinity worse in the Baltic sea - are there problems in brackish waters?

Line 522: "upper 50 m of the water column" did you mean 500m as in the plot?

Line 540 - should be Figure 14?

figure 15(b) - are units km?

Line 545: please refer to table 3 for the RMS numbers too.

Table 3: Is this R or Rˆ2 you are listing?

Line 546/546: "likely representing the delay in the analysis system in adjusting to SLA errors". Could it not be due to the non-stationary location of the front/eddies? And looking only at observations with a 10 day return period you are not capturing the evolution of the model between overpasses?

Line 553 - Figure 15 -> Figure 14. Also subsequent references to Fig 16 -> Fig 15.

Figure 15: Caption is for Figure 14

Figure 16 not present but referenced.

Line 622 "OPP" -> "YOPP"?
* * *

---

## Referee Comment (RC2) · Anonymous Referee #2 · 9 Nov 2020

General comments The article presents the new configuration of the RIOPSv2 Arctic forecasting system, which - compared to its predecessor - counts two novelties related to the assimilation of SST data and an advanced tidal filtering for the assimilation of SLA. The new system is presented in many details, including illustrations of the multi-variate and anisotropic spatial background covariance from the ensemble, a hardcore mathematical derivation of the harmonic analysis used for the SLA assimilation and a comparison of the system results to its mother system, the global non-tidal, coarse-resolution GIOPS. However, the paper does not evaluate any of the improvements from the beginning to the end, which limits the impact of the paper. This applies particularly to the time-dependent harmonic analysis, which represents a significant effort

but which results are frustratingly terse. Is the filter robust? How does it compare to other tidal filters that are not designed for ice-covered areas? Even though the article does not perform a clean assessment of the advanced harmonic analysis against a more rudimentary filter, the paper is overall of high standards and worthy of publication as a description of an operational system. The main barriers to appreciating it are the length of the paper and the tendency to accumulate distracting topics that do not help those who may be tempted to reproduce the results. I would recommend the paper is published under the condition that the authors focus on the novel topic of the paper - the harmonic analysis - and provide more ample evidence that the approach is worth the effort.

A secondary innovation of the paper is the smoothing of model SST fields before assimilation of high-resolution SST, which visually improves the innovation field. I believe that this smoothing does not interfere with the effect of the harmonic analysis and that it tends to make the comparison of RIOPS to GIOPS more relevant, so this part should remain in the paper.

The multivariate and anisotropic structures of the ensemble covariance matrix, however, are not a specific novelty of the present paper but are common to all applications of SAM and other ensemble-based techniques. Since these are not used to explain the results, I would shorten that part to a few sentences. Figures 3 to 6 are also smashing graphics, but they are of little relevance to the rest of the paper. I would recommend removing them (maybe leave one) and their description to shorten the paper.

The description of the assimilation cycle would deserve some clarification and one figure has gone missing. Otherwise, the paper is very well written and makes an interesting read.

specific comments - The title of the paper is too generic to indicate its actual contents. There is no way that anyone interested in the harmonic analysis in seasonal ice-covered waters would track it back to this paper unless the keywords "tidal" and

"observation operator" are in the title.

- l39: Salinity biases of 0.3 - 0.4 psu sound extremely good, so it should be noted that these are averaged over the top 500 m.

- The introduction does not cite any previous attempts to assimilate SLA data in a tidal-driven model. Discarding the papers dedicated to the estimation of tidal parameters, a reference to the tidal GOFS v3.1 from NRL and the more rudimentary method by Xie et al. (2011) could indicate what is available.

Xie, J., Counillon, F., Zhu, J., and Bertino, L.: An eddy resolving tidal-driven model of the South China Sea assimilating along-track SLA data using the EnOI, Ocean Sci., 7, 609–627, https://doi.org/10.5194/os-7-609-2011, 2011.

- l130 Has Paquin et al. (in prep.) become accessible in the meantime?

- The RIOPS v1.3 has not been used in the whole paper, so the description of the old system (including Table 1) should be removed to shorten the paper.

- l158 if the SST is not cycled with the assimilation, how is it assimilated then?

- l170 it took me a long time to understand the 3 assimilation cycles in Figure 2. What is assimilated in the RR cycle? Are the SST and sea ice concentrations not assimilated in the RD and RR cycles? Then, the authors should specify that the first "R" stands for "Regional".

- l176 Pham et al. 1998 describe the evolutive basis of the SEEK filter, please specify that you use a fixed basis here.

- l184 Talagrand's (1998) adaptivity scheme is not common knowledge. Is it following the criterion that the cost function should remain superior to half the number of observations?

- l189 contradicts l159 where the bias correction is only planned.

- l207 is the SST projected in the vertical or nudged at the surface?

- l277 should refer to Figure 7, not 8.

- l310 if C contains the phase, it should be dependent on k. Please note it Ck then.

- l317 from Eq 3a to 3b, only the left-hand term of the product has been conjugated. My maths are buried too deep in my brain to remember why. Please explain briefly.

- l318 Wnm seems to be a temporal covariance matrix. If it is diagonal, this means that the tidal residuals are assumed to be white noise, can you confirm?

- l318 why bother with two indices nm if the W matrix is diagonal?

- l331 if C is depending on the frequency, can it be cancelled?

- l345 The restoring time length appears discretely in parenthesis. I believe this is the only arbitrary parameter of the method please explain the choice of 30 days.

- l350 Can you shorten this sentence using the (m-1)th, or (m-1)st, time step? I find the use of the prime instead of -1 cumbersome.

- l355 The weights are decreasing exponentially. This should be stated explicitly.

- l365 I am missing an illustration of the tidal filter weights, at one sample point in winter and in summer, which could be compared to a more common tidal filter. See for example a few convolutions below, some one them can work one-sided, using data from the past only: https://www.sonel.org/Filters-for-the-daily-mean-sea.html

- l422 The typical amplitude of the Msf constituent would be useful to know here. Is it worth including it at all if its removal is so problematic?

- l462 Again, information about RIOPSv1 should be removed as the system is not used in the paper.

- l535 Are there sufficient profiles in the Arctic for a robust bias correction? It seems the positive impact is only noticed in the Southern part of the domain.

- l585 The MDT in the central Arctic is coming from a different system, the GLORYS reanalysis, which is prone to inconsistencies. Did GLORYS assimilate ITP profiles for example?

- l635 This paragraph is only loosely related to the rest of the paper, is it necessary?

- l962-963. Too long sentence, I cannot follow the point.

- l1036. Is this figure a snapshot? Is it taken in the summer or winter? This could affect the amplitude of the inverse barometer component.

technical corrections - l341 Missing "a" before diagonal - l514 Missing "the" before Arctic Ocean. - l622 Has the OPP acronym been defined before? - l974 the element IS involving THE sine dimension - Figures 9 to 13 are too small, it is hard to see what happens in the North Labrador Sea - Figure 14 (deep biases) is missing but the caption remained, so the captions are shifted thereafter.

---

## Author Comment (AC2) · 23 Dec 2020

**Response to Reviewer 1**

Text from reviewers is in black italics with responses in blue.

This paper describes and evaluates the high resolution CCMEP analysis which spans the coastline of Canada and the Arctic. The model and assimilation methodology are introduced in relation to the previous regional system. The performance of the system is assessed by an evaluation of background departure statistics against sea level anomaly and in-situ observations, and compared against the Global Ice Ocean Prediction System.

Firstly let me apologize to the authors for the delay to my review and thank the editor for allowing me sufficient time to complete it. I am glad to be able to have thoroughly read the paper and I feel it very well describes the operational system. Overall I am sure this paper is perfectly suited to GMD and should be published subject to some minor corrections/additions.

*I* will first give my more general comments, and subsequently give the itemized comments *I* had whilst reading through the paper.

**General comments:**

The paper is very well written and was enjoyable to read. For a description of an operational analysis system the authors have the balance just about right in terms of the detailed description within this paper and referencing previous work.

**Thank you very much.**

In some areas I was not clear whether those references would suffice or if they were differences between the previous system and RIOPSv2. For example the boundary conditions used for the regional system are not well highlighted (I assume they come from GIOPS but this isn't explicitly stated?). Also when the change to model levels was mentioned it highlighted that the vertical domain was not well described and may lead to confusion so a sentence or two would go a long way to help the reader here.

Additional details have been added regarding the open boundary conditions and the statement regarding the model levels was modified to improve clarity.

The section on the online harmonic analysis was far more in depth than the others. I am not familiar with such methods and therefore not sure where the methodology from the literature finishes and the new science (I think related to the sliding window) starts. If this could be highlighted it would be a useful addition.

In response to this comment and a request by Reviewer 2 to include a greater discussion of the performance of the online harmonic analysis scheme several changes have been made. A paragraph has been added to the Introduction to provide better context, including previous studies that assimilate SLA in an ocean model with tidal variations. An additional figure and discussion of the performance of online harmonic analysis filter compared to the well-known T\_tide package has been added to Section 3.4.4

More generally what is not discussed is the mismatch between a model that contains tides (needed for the coastal applications) and SLA observations which do not contain tidal signals. However the raw altimetry data would necessarily contain tidal signals, so it would benefit the reader to have a short section on SLA observations and why you aren't using lower level SLA observations.

Assimilating lower level SLA observations that include tides would only be suitable if the intend was to improve the representation of the tides themselves. Since the tidal errors are mainly stationary, this can be treated in advance to separate the tidal and non-tidal signals present in the altimetry data. This also allows use of "standard" altimetry products that include the tidal filtering along with the other processing steps, such as the

dynamic atmospheric correction and long-wave error filtering. A comment to this effect has been added to Section 3.3.3.

I was hoping to see more about sea ice performance but the final paragraph is sufficient to let the reader know why it is not shown. I wonder if this could be expanded upon to hint at how a future system might look to improve sea ice predictions?

As noted by Reviewer 2, sea ice forecasting is outside the main focus of this paper. As such, we don't feel it is appropriate to expand upon this topic in this manuscript. There are other papers that focus directly on sea ice forecasting in RIOPS (e.g. Lemieux et al. 2016; Chikhar et al., 2019).

Finally I would ask that the authors look at the colour scales they are using in their plots. I think a lot of the structures we see are artifacts of the jet colour scheme and the use of perceptually uniform colour scales would remove such artifacts and make the paper more accessible to colour blind readers.

Thank you for the suggestion. In the revised manuscript, Figs. 6-11 have been reproduced using perceptually uniform colour scales. Conclusions and comments in the text regarding these figures remains unchanged.

Detailed comments:

Line 126: I am unfamiliar with these tidal constituents - is this detail necessary and if so should such parameters be described?

Constituents specified are typical and well known within tidal modelling community. However a different number of constituents are used in different contexts. Providing the list of constituents here improves reproducibility of the results.

Line 137: Please comment on the vertical domain - "deep layers (from 500m" suggests these are the deepest layers in your domain, but I would expect the domain to go down to \_4000m in the arctic.

Thank you for pointing this out. Indeed, the wording is confusing. The sentence has been modified to clarify that the increased resolution is focused between 250 m and 500 m.

Line 170: This has strong similarities to the ECMWF workflow shown in Browne et al. 2019 Figure 3. ::: Browne, P. A., de Rosnay, P., Zuo, H., Bennett, A., & Dawson, A. (2019). Weakly Coupled Ocean-Atmosphere Data Assimilation in the ECMWF NWP System. Remote Sensing, 11(234), 1–24. https://doi.org/10.3390/rs11030234

We agree with the reviewer that there are strong similarities between the approaches. A comment to this effect has been added to the text.

Lines 209-211: "Another modification required for coupled forecasts was to use 24-h averaged short and long-wave radiation fields to force NEMO-CICE during the analysis cycles such that there is very little diurnal warming present in the ocean analysis". Please can you elaborate on this. Is the system designed to have a strongly damped diurnal cycle in the analysis, or are there reasons for which this was found to be necessary?

The following text has been added to clarify:

"Damping diurnal SST variations in the analysis fields was also found to limit initialization shock in coupled forecasts as the atmospheric analysis was produced using a foundation SST product (Smith et al., 2018)."

Line 215: Smith et al (2015) should be 2016? Or a different paper?

Thanks for pointing out this error. The article was available online in 2015. All references to this article have now been changed to 2016 to avoid confusion.

Line 228: Can you clarify - does this mean you apply the increment from the current cycle to the next cycle (in the first 24 hours)?

Yes, that's correct. As such, the first day of the 7-day IAU run has two increments applied: an increment with a linearly decreasing ramp from the previous cycle, and an increment with a linearly increasing ramp from the current cycle. This helps to make a smooth transition from one cycle to the next.

Figures 3-6: Subplot titles are missing commas, i.e. <SST(r)SST(r+\Delta r)> should be <SST(r),SST(r+\Delta r)>

Perhaps this is a case of different conventions. It is quite common to express an expectation of a product (say A and B) as <AB>. Also, as noted below, it is also quite difficult for us to reproduce these figures as it is computationally time consuming.

Figure 3 - star is not visible, please can you add it on top or produce the image at a higher resolution? In fact maybe just change the text to make it clear the point you are considering is the bottom left corner of the magenta box?

**The star has been indicated more clearly.**

Figure 3 - this is the first time you mention localisation. Please can you elaborate on why you choose the region that you do? Is the localisation a tensor product of horizontal and vertical localisation functions, or is there no localisation in the vertical? Is it a hard cut off or is it smoothly reduced to zero at the boundary of the "bubble" by a Gaussian or Gaspari-Cohn function? (I suspect you will have a reference for this which would be good, but also nice to know if you have had to make any changes here for RIOPSv2).

There is no localization in the vertical. The horizontal localization is applied using a Gaussian function with the the length scale indicated in the figure. The figure caption has been modified to clarify this point. No changes have been made to the localization.

Figures 3-11,13,14: If possible please replot using a diverging, perceptually uniform colour scale. For instance in Figure 3, <SST(r)T(r+\Delta r)>, the current choice of colourscale appears to show a clear change in correlation around layer 10 whereas I think this is an artifact of the rainbow colour scale. Thyng, Kristen M., et al. "True colors of oceanography: Guidelines for effective and accurate colormap selection." Oceanography 29.3 (2016): 9-13. https://doi.org/10.5670/oceanog.2016.66 Hawkins, Ed. "Scrap rainbow colour scales." Nature 519.7543 (2015): 291-291. https://doi.org/10.1038/519291d

The colour scale has been changed to a perceptually uniform colour scale for Figs. 6-11, as these figures include the main quantitative comparison presented in this paper. As the other figures are used for qualitative comparison we feel the use of the jet colour scale does affect the scientific interpretation of the results. Reproducing Fig. 3 would be quite difficult as the calculations involved are extremely time consuming. The colours are used to provide a qualitative indication of the covariance scales, and as such we feel they are suitable.

Line 278: "model trial field" is not well defined. I suggest replacing with something like "raw model equivalent field"

Text modified as suggested.

Figure 7: Please relabel "CMC" -> "CCMEP" and "Trial" -> "RIOPSv2"

Figure modified as suggested.

Line 299: ", the use of a rotation" -> ", and the use of a rotation"

Modified as suggested (but with the comma removed).

Line 306: Are you using Einstein notation here so there is an implicit summation over the repeated index k? This should be noted. Oh I now see it introduced on line 320 - is it used before or only from line 320?

Text clarified to indicate that Einstein notation used throughout.

Line 311 - the definition of \omega\_k is unclear. please look carefully at the wording you use here.

Changed "at" to "is".

Line 315: Is this a minimisation over only Xk?

Yes. To clarify, we changed the equation as follows:

$$J(X^{k}) = \frac{1}{2}(A^{n} - H^{n})^{*}W_{nm}(A^{m} - H^{m})$$

Line 338: Prime symbol is not consistent - MS Word issue?

Thank you for noticing this. Text corrected.

Line 350: "According to the definition of the sliding window weight". Where is this defined? Eqn 3a? Also have you defined explicitly what "its property of normalisation" is?

Paragraph rewritten. Figure added in response to Reviewer 2 to provide more detail regarding the sliding window weight.

Figure 8: Can you clarify, possibly in the caption, that (a) = (b) - (c) - (d)? Again, diverging colour scales would be much better here. In fact the total SSH field (b) is confusing: "Panel (b) shows the instantaneous model SSH field prior to any treatment." but its range is (-1,1). Has this already had the MDT removed? I personally am more familiar with considering the MDT for SLA assimilation when using processed SLA observations that have tidal signals removed, so I would like to see precisely where the MDT is required in your methodology.

The caption has been modified as requested and the figure has been reproduced using diverging colour scales.

The SSH field is the full field, prior to removing the MDT. The MDT is removed as is usually done as part of the observation operator. This is the same for a model without any tides. Perhaps there is some confusion regarding the SLA observations assimilated, as the SLA observations used here are the commonly-used AVISO product that have the tides removed.

Line 477/478. Regarding the Gulf stream errors, would you expect the SEEK filter methodology to be able to effectively constrain a region which such variability given the error covariances are coming solely from a climatology and therefore may be far too smooth there? It is nice to see the errors reduced in RIOPSv2.

This is an important point. The covariances do not come from a climatology, but rather are constructed from sub-monthly anomalies from a 10-year model run. As such, each anomaly (or error mode) is at roughly the same resolution as the model (apart from SST for which there is additional filtering as described in the text. The approach was developed with the intent to constrain the mesoscale variability (See Lellouche et al., 2013 and references therein). That being said, GIOPS is only of eddy-permitting resolution and would resolve less well the structures in the Gulf Stream region, whereas RIOPS resolves these features better. The text has been modified to clarify this point.

Line 485: The difference in errors in the Laptev sea is intriging. Could you produce a comparison plot of the two different MDTs used?

Both RIOPS and GIOPS use the same MDT field. The confusion is probably due to the word "field" being erroneously used in the plural. This has been corrected in the revised manuscript.

Line 488: We should see errors here in salinity too if that is the case? [Refer to my later comment on Figures 11-13]

Unfortunately, there aren't any in situ observations in the areas for which we see the largest biases (Hudson's Bay, northern Laptev Sea, mouth of Lena River). When activating the bias correction scheme we did note that the mean SLA innovations increased suggesting that water mass errors do contribute to this signal.

Line 489-491: Surely this is entirely due to ice cover - polar orbiting satellites should have much richer coverage in the arctic due to their much more regular return period (if you exclude the polar data gap).

While the reduced number of observations for Sentinel3, Altika and Cryosat2 are due to sea ice cover, observations from Jason3 are also included, which do not cover the Arctic Ocean. As such we feel the statement in the manuscript is correct: "...due to satellite orbits and ice coverage, many fewer observations are present over the Arctic Ocean..."

Section 4.2.2/Figure 10. Given the broadly similar structures and different scales between Gulf stream errors and other regions it would be nice to see a difference plot and/or a normalised difference plot of 10(a) and (b). This might be too difficult to produce in a reasonable amount of time/effort, but may prove enlightening.

Thank you for the suggestion, but we find using a normalized difference plot over-emphasizes small differences, whereas we prefer to focus on the main patterns and areas of error.

Figures 11-13: They appear to be missing the polar sector from 45E to 180E. Is this a plotting error? Are there no in-situ obs in the Hudson bay? I was hoping to see salinity departures which would corroborate interpretation of errors around the mouth of the Lena river that you associated to the fresh water fluxes.

We fully agree with the reviewer that in situ observations in Hudson Bay, at the mouth of the Lena River and more generally throughout the Arctic would be of significant value, in particular for helping to corrected diagnose and improve SLA errors. Unfortunately, there are no observations available in these specific regions in the dataset we use. The availability of in situ data in the Arctic is a significant issue (for a review see Smith et al., 2019).

figure 11 - Salinity worse in the Baltic sea - are there problems in brackish waters?

The problem appears quite localized and different than what is seen, for example, in the Gulf of St. Lawrence. As such, we don't feel it's a generalized problem but rather likely associated with errors in freshwater runoff applied in the Baltic Sea.

Line 522: "upper 50 m of the water column" did you mean 500m as in the plot?

Thank you for pointing this out. Figure 12 is incorrectly referenced in this context. The error is indeed localized in the upper 50 m as indicated in the text. The text has been modified to indicate that this is "not shown" (rather than referencing Fig. 12).

Line 540 - should be Figure 14?

Yes. Thank you for catching this oversight.

figure 15(b) - are units km?

Yes. Units for other subplots have also been added.

Line 545: please refer to table 3 for the RMS numbers too.

A reference to Table 3 was added after the comment regarding the RMS SLA innovations.

Table 3: Is this R or R2 you are listing?

As indicated in the Table caption, the correlation values (r) are provided, not the proportion of explained variance  $(r^2)$ .

Line 546/546: "likely representing the delay in the analysis system in adjusting to SLA errors". Could it not be due to the non-stationary location of the front/eddies? And looking only at observations with a 10 day return period you are not capturing the evolution of the model between overpasses?

We agree with the reviewer that there may be others causes of the differences noted in the figure. In addition to the theory proposed by the reviewer, they may also be associated with anomalies in the observations. We have removed this speculation from the manuscript.

Line 553 - Figure 15 -> Figure 14. Also subsequent references to Fig 16 -> Fig 15. Figure 15: Caption is for Figure 14 Figure 16 not present but referenced.

Our sincerest apologies for the oversight related to the missing caption and incorrect figure references and any confusion it may have caused in reviewing our manuscript. In response to comments by Reviewer 2, additional figures have been removed and all figures numbers and references have been updated.

Line 622 "OPP" -> "YOPP"?

No. OPP is correct. The acronym has been expanded in the revised manuscript as Ocean Protection Plan.

---

## Author Comment (AC3) · 23 Dec 2020

**Response to Reviewer 2**

Text from reviewers is in black italics with responses in blue.

**General comments**

*The article presents the new configuration of the RIOPSv2 Arctic forecasting system, which - compared to its predecessor - counts two novelties related to the assimilation of SST data and an advanced tidal filtering for the assimilation of SLA.*

This is not quite right, the previous version of RIOPS did not have its own data assimilation system but rather used a spectral nudging approach to constrain temperature and salinity fields towards GIOPS analyses. As such, the introduction of the data assimilation system for RIOPS is perhaps the main novelty. The online tidal filter is also novel. The manner in which we assimilate SST data is not new, but has not previously been published.

*The new system is presented in many details, including illustrations of the multivariate and anisotropic spatial background covariance from the ensemble, a hardcore mathematical derivation of the harmonic analysis used for the SLA assimilation and a comparison of the system results to its mother system, the global non-tidal, coarse resolution GIOPS. However, the paper does not evaluate any of the improvements from the beginning to the end, which limits the impact of the paper.*

As this is the first analysis system for RIOPS (and the first pan-Canadian regional analysis system), the main objective of the paper is simply to document the RIOPSv2 system and evaluate the analysis quality as compared to GIOPS (this provides a proxy for comparing to RIOPSv1, since RIOPSv1 was nudged toward GIOPS). Specific "beginning to end" evaluations may build upon this effort.

*This applies particularly to the time-dependent harmonic analysis, which represents a significant effort but which results are frustratingly terse. Is the filter robust? How does it compare to other tidal filters that are not designed for ice-covered areas?*

We agree with the reviewer that the harmonic analysis merits a clear demonstration. A figure has been added to demonstrate the robustness of the online filter as compared to the commonly used "T-tide" filter with a discussion provided in Section 3.4.4.

*Even though the article does not perform a clean assessment of the advanced harmonic analysis against a more rudimentary filter, the paper is overall of high standards and worthy of publication as a description of an operational system. The main barriers to appreciating it are the length of the paper and the tendency to accumulate distracting topics that do not help those who may be tempted to reproduce the results. I would recommend the paper is published under the condition that the authors focus on the novel topic of the paper - the harmonic analysis - and provide more ample evidence that the approach is worth the effort.*

As noted above, a figure and discussion has been added to focus more on the harmonic analysis. To reduce the "distracting topics", the number of figures included to highlight the covariance structures has been reduced. We do feel that showing these covariance structures and the filtering used in the SST observation operator are essential components of the RIOPSv2 system that warrant inclusion to adequately document how the analysis system is constructed.

*A secondary innovation of the paper is the smoothing of model SST fields before assimilation of high-resolution SST, which visually improves the innovation field. I believe that this smoothing does not interfere with the effect of the harmonic analysis and that it tends to make the comparison of RIOPS to GIOPS more relevant, so this part should remain in the paper.*

Thank you. We agree.

*The multivariate and anisotropic structures of the ensemble covariance matrix, however, are not a specific novelty of the present paper but are common to all applications of SAM and other ensemble-based techniques.*

*Since these are not used to explain the results, I would shorten that part to a few sentences. Figures 3 to 6 are also smashing graphics, but they are of little relevance to the rest of the paper. I would recommend removing them (maybe leave one) and their description to shorten the paper.*

As noted above, only one figure showing covariance structures has been included in the revised manuscript as suggested. The text in this section has also been shortened accordingly. The technical details describing the construction of the error modes has been left in the manuscript as this is an important component of the system.

*The description of the assimilation cycle would deserve some clarification and one figure has gone missing.*

The description of the assimilation cycle has been improved as you suggest below (specific suggestions) and the issue with the figure captions has been corrected. Our apologies for this oversight.

*Otherwise, the paper is very well written and makes an interesting read.*

Thank you very much for your kind comments and for taking the time to review the manuscript.

**Specific comments**
*- The title of the paper is too generic to indicate its actual contents.*
*There is no way that anyone interested in the harmonic analysis in seasonal*
*ice-covered waters would track it back to this paper unless the keywords "tidal" and "observation operator" are in the title.*
Thank you for the suggestion. We have changed the title to : "The Regional Ice Ocean Prediction System v2: A pan-Canadian ocean analysis system using an online tidal harmonic analysis"

*- l39: Salinity biases of 0.3 - 0.4 psu sound extremely good, so it should be noted that these are averaged over the top 500 m.*

Abstract modified as suggested.

*- The introduction does not cite any previous attempts to assimilate SLA data in a tidaldriven model. Discarding the papers dedicated to the estimation of tidal parameters, a reference to the tidal GOFS v3.1 from NRL and the more rudimentary method by Xie et al. (2011) could indicate what is available.*
*Xie, J., Counillon, F., Zhu, J., and Bertino, L.: An eddy resolving tidal-driven model of the South China Sea assimilating along-track SLA data using the EnOI, Ocean Sci., 7, 609–627, https://doi.org/10.5194/os-7-609-2011, 2011.*

A paragraph has been added to the introduction to discuss assimilation of SLA data in ocean models that include tidal variations, including a mention of the Xie et al. (2011) study. We were unable to identify a suitable reference for the GOFSv3.5 system (GOFSv3.1 does not contain tides). We even contacted the scientists responsible for the development of GOFS.

*- l130 Has Paquin et al. (in prep.) become accessible in the meantime?*

No. This reference has been removed from the revised manuscript.

*- The RIOPS v1.3 has not been used in the whole paper, so the description of the old system (including Table 1) should be removed to shorten the paper.*

It is a reference to indicate how the new system differs from the previous operational version. As such it a useful reference to aid in reproducibility.

*- l158 if the SST is not cycled with the assimilation, how is it assimilated then?*

As described in Section 3.3.4, the gridded SST analysis produced by CCMEP is assimilated into the SAM2 system used by RIOPS. The use of a gridded SST analysis products avoids the need to address various issues associated with the assimilation of SST data (e.g. sensor biases, data density) and provides greater

consistency between surface conditions from the ocean model and those used in producing the atmospheric forcing, thereby reducing unphysical surface fluxes (see Smith et al., 2018). This approach is fairly standard and has been used by various studies in the past, e.g. Lellouche et al. (2013, 2018).

*- l170 it took me a long time to understand the 3 assimilation cycles in Figure 2. What is assimilated in the RR cycle? Are the SST and sea ice concentrations not assimilated in the RD and RR cycles? Then, the authors should specify that the first "R" stands for "Regional".*

Additional detail has been provided to the text describing Fig. 2 to clarify. In particular, it is noted that the "R" in RD, RR and RU stands for RIOPS. Also, both RD and RR assimilate all available observations, albeit RD has a longer cutoff and thus includes a greater number of in situ and SLA observations.

*- l176 Pham et al. 1998 describe the evolutive basis of the SEEK filter, please specify that you use a fixed basis here.*

Done.

*- l184 Talagrand's (1998) adaptivity scheme is not common knowledge. Is it following the criterion that the cost function should remain superior to half the number of observations?*

The idea is that the variance of the innovations should be equal to the variance of the background error plus the variance of the observations error.

*- l189 contradicts l159 where the bias correction is only planned.*

Line 159 indicates that the inclusion of the bias correction (evaluated in this manuscript) is planned to be implemented operationally in fall 2021. The sentence has been reworded to clarify. The main point is that the version of RIOPS evaluated here (v2.1) is planned for operational implementation in fall 2021.

*- l207 is the SST projected in the vertical or nudged at the surface?*

The gridded CCMEP OI SST product is assimilated together with in situ observations of temperature and salinity and SLA anomaly observations using the SEEK approach. As such there is no need to project the SST in the vertical, nor to nudge it at the surface. Multi-variate corrections are produced based on the covariances in the background error modes.

*- l277 should refer to Figure 7, not 8.*

Corrected.

*- l310 if C contains the phase, it should be dependent on k. Please note it Ck then.*

Thanks, we denote C as $C_k$, then equation (2) becomes as

$$E_k^n = C_k exp(i\omega_k \tau n) \qquad (2)$$

*- l317 from Eq 3a to 3b, only the left-hand term of the product has been conjugated. My maths are buried too deep in my brain to remember why. Please explain briefly.*

Thanks for pointing out this error. Equation (3a) need be corrected as following:

$$J = \frac{1}{2}(A^n - H^n)^* W_{nm}(A^m - H^m) , \qquad (3a)$$

*- l318 Wnm seems to be a temporal covariance matrix. If it is diagonal, this means that the tidal residuals are assumed to be white noise, can you confirm?*

Yes, we assume tidal residuals are uncorrelated in time and represent $W_{nm}$ as a diagonal matrix that specifies the time weights used in the least-square fit. Ignoring the time correlation allows us to represent $W_{nm}$ as a diagonal matrix rather than a vector in order to use the Einstein Summation Convention.

*- l318 why bother with two indices nm if the W matrix is diagonal?*

As explained above

- l331 if C is depending on the frequency, can it be cancelled?

Yes, it can be cancelled because $C_k$ is time-independent (independent of time step n). This can be shown by denoting $F_j^n = \exp(i\omega_j \tau n)$ and rewriting Eq. (2) as $E_k^n = D_k^j F_j^n$, where $D_k^j$ is a diagonal matrix with $C_k$ as its $k^{th}$ diagonal element. Substituting this into the definition of matrix B and vector Y, we have

$$B_{kj} = D^*{}_k^{k'} F^*{}_{k'}^n W_{nm} F^m{}_{j'} D^{j'}{}_j = D^*{}_k^{k'} G_{k'j'} D^{j'}{}_j$$

$$Y_k = D^*{}_k^{k'} F^*{}_{k'}^n W_{nm} H^m = D^*{}_k^{k'} U_{k'}$$

By taking the inverse of the B matrix by the formula $(D^*GD)^{-1} = D^{-1}G^{-1}D^{*-1}$, and substituting the above equations into Eq. (6a), matrix D (and thus $C_k$) is cancelled. This means that the final solution of $A_n$ is independent of $C_k$.

*- l345 The restoring time length appears discretely in parenthesis. I believe this is the only arbitrary parameter of the method please explain the choice of 30 days.*

Thank you for pointing out this omission. The following text has been added in Section 3.4.4 :
"The other free parameter in the sliding window approach is the time scale used in weighting function. Here a value of 30 days is used. This value must be large enough to permit an accurate fit of the different tidal constituents. Using a longer value reduces the ability of the system to adapt to seasonal changes in tidal variability."

- l350 Can you shorten this sentence using the (m-1)th, or (m-1)st, time step? I find the use of the prime instead of -1 cumbersome.

The use of the prime provides a straightforward way to indicate quantities valid at the previous time step. This is intended to aid in producing a numerical code based on this algorithm.

- l355 The weights are decreasing exponentially. This should be stated explicitly.

Done.

- l365 I am missing an illustration of the tidal filter weights, at one sample point in winter and in summer, which could be compared to a more common tidal filter. See for example a few convolutions below, some one them can work one-sided, using data from the past only: https://www.sonel.org/Filters-for-the-daily-mean-sea.html

As pointed out above, the tidal filter weights are equivalent to an exponential function based on the specified timescale (30 days). These weights have been plotted in the Fig. R1 below. Here, the weighting function has been plotted along with an exponential function to illustrate their equivalence. Note that there is no summer or winter dependence, nor any spatial variation in the value of the filter weights. They are strictly a function of the specified timescale.

[Figure]

*Fig. R1: Comparison of the tidal filter weighting function with an exponential function. Both functions are plotted as a function of days, using the time step from RIOPS (i.e. 288 steps per day) and a restoring timescale of 30 days.*

For comparison, we've also plotted a few common tidal filters in Fig. R2. We performed a harmonic analysis at several points using these three filters. While the two-sided functions provide a roughly equivalent result to that of T_tide and the online filter, the one-sided function (required for real-time operational use) results in a significant phase lag and large residuals.

[Figure]

*Fig. R2: Normalized tidal filter weights for a few common filters (Munk "tide killer", Godin and Demerliac) with the value of K in hours. Filters were reproduced using information from the Sonel.org website based on information from Bessero (1985).*

*- l432 The typical amplitude of the Msf constituent would be useful to know here. Is it worth including it at all if its removal is so problematic?*

The Msf is simply used here as an example of a long-period tidal constituent. The amplitude of the Msf constituent (and other long-period constituents) varies considerably over the domain, but is generally much smaller than the principle semi-diurnal and diurnal constituents. Indeed, they are not worth including in the online harmonic analysis given their amplitude and the associated difficulties in removing them. These sentences have been removed to avoid confusion.

*- l462 Again, information about RIOPSv1 should be removed as the system is not used in the paper.*

The reference here to RIOPSv1 is essential to explain our evaluation approach. One of the main objectives of the paper is to provide an evaluation of the analysis system implemented in RIOPSv2. Since RIOPSv1 was nudged toward GIOPS, we use GIOPS as a proxy for RIOPSv1.

*- l535 Are there sufficient profiles in the Arctic for a robust bias correction? It seems the positive impact is only noticed in the Southern part of the domain.*

Indeed, the number of profiles in the Arctic is smaller than in other regions (Fig. R3). However, in some areas, such as the Beaufort Sea they show a consistent pattern revealing positive salinity biases. Moreover, this number of profiles is greater than is usually present due in part to the additional profiles deployed as part of the Year of Polar Prediction (as discussed in the manuscript).

[Figure]

Fig. R3: Number of profiles used in innovation statistics (Fig. 10) of in situ salinity over the upper 500 m for the period 01-Jan-2016 to 31-Dec-2018.

*- l585 The MDT in the central Arctic is coming from a different system, the GLORYS reanalysis, which is prone to inconsistencies. Did GLORYS assimilate ITP profiles for example?*

As described in Section 3.1, the MDT used is a hybrid product that uses the CNES-CLS 2013 product as a base with modifications based on mean increments from the GLORYS reanalysis (as described by Lellouche et al., 2018). Owing to the small number of data in the Arctic, especially over the earlier period of the GLORYS reanalysis, the modifications to the CNES-CLS 2013 MDT in the Arctic are quite minor (nor do we believe that GLORYS assimilated ITPs). The main changes are found near Indonesia, the Red Sea and Meditarrenean Sea (Lellouche et al., 2018).

*- l635 This paragraph is only loosely related to the rest of the paper, is it necessary?*

This paragraph was included as RIOPS is often used for sea ice prediction and so a comment regarding the impact on sea ice related applications is relevant (indeed Reviewer 1 requested that we expand on this!).

*- l962-963. Too long sentence, I cannot follow the point.*

The sentence has been simplified as follows: "This means that the 2K+1 dimensional real vector space is a complete invariant subspace when operating under S"

*- l1036. Is this figure a snapshot? Is it taken in the summer or winter? This could affect the amplitude of the inverse barometer component.*

Yes, the figure is a snapshot valid for 31-Dec-2015. The date used as been indicated in the caption for the revised figure. While the amplitude of the inverse barometer varies based on the synoptic situation, it remains small compared to the amplitude of the tidal signal.

***Technical corrections***
*- l341 Missing "a" before diagonal*
Corrected.

*- l514 Missing "the" before Arctic Ocean.*
Corrected.

*- l622 Has the OPP acronym been defined before?*
OPP Acronym written out in full in revised manuscript (Ocean Protection Plan).

*- l974 the element IS involving THE sine dimension*
Corrected.

*- Figures 9 to 13 are too small, it is hard to see what happens in the North Labrador Sea*
If you zoom in (increase magnification) it is possible to see the details in the Labrador Sea quite clearly. We`ll ensure that the figure quality is sufficient to see these features clearly during the proofing stage of the paper.

*- Figure 14 (deep biases) is missing but the caption remained, so the captions are shifted thereafter.*
Our apologies for this error and any confusion it caused for the review. Figure captions and references have been corrected in the revised manuscript.

**References:**
Bessero G. (1985). Marées. SHOM, Fascicule 2, chap. IX à XV.

Chikhar, K., Lemieux, J.F., Dupont, F., Roy, F., Smith, G.C., Brady, M., Howell, S.E. and Beaini, R.: Sensitivity of Ice Drift to Form Drag and Ice Strength Parameterization in a Coupled Ice–Ocean Model. Atmos. Ocean, pp.1-21, 2019.

Lellouche, J. M., Le Galloudec, O., Drévillon, M., Régnier, C., Greiner, E., Garric, G., Ferry, N., Desportes, C., Testut, C. E., Bricaud, C. and Bourdallé-Badie, R. : Evaluation of global monitoring and forecasting systems at Mercator Océan, Ocean Sci., 9(1), 57, 2013.

Lemieux, J.-F., Dupont, F., Blain, P., Roy, F., Smith, G.C. and Flato, G.M.: Improving the simulation of landfast ice by combining tensile strength and a parameterization for grounded ridges, J. Geophys. Res., 121(10), 7354-7368, 2016.

Lellouche, J. M., Greiner, E., Le Galloudec, O., Garric, G., Regnier, C., Drevillon, M., Benkiran, M., Testut, C. E., Bourdalle-Badie, R., Gasparin, F. and Hernandez, O.: Recent updates to the Copernicus Marine Service global ocean monitoring and forecasting real-time 1⁄12° high-resolution system. Ocean Sci., 14(5), 1093-1126, 2018.

Smith, G. C., Bélanger, J. M., Roy, F., Pellerin, P., Ritchie, H., Onu, K., Roch, M., Zadra, A., Surcel Colan, D., Winter, B. and Fontecilla, J. S.: Impact of Coupling with an Ice-Ocean Model on Global Medium-Range NWP Forecast Skill. Mon. Wea. Rev., 146, 1157-1180, doi: 10.1175/MWR-D-17-0157.1, 2018.

Smith, G. C., Allard, R., Babin, M., Bertino, L., Chevallier, M., Corlett, G., Crout, J., Davidson, F., Delille, B., Gille, S. T., Hebert, D., Hyder, P., Intrieri, J., Lagunas, J., Larnicol, G., Kaminski, T., Kater, B., Kauker, F., Marec, C., Mazloff, M., Metzger, E. J., Mordy, C., O'Carroll, A., Olsen, S. M., Phelps, M., Posey, P., Prandi, P., Rehm, E., Reid, P., Rigor, I., Sandven, S., Shupe, M., Swart, S., Smedstad, O. M., Solomon, A., Storto, A., Thibaut, P., Toole, J., Wood, K., Xie, J., Yang, Q. and the WWRP PPP Steering Group: Polar Ocean Observations: A Critical Gap in the Observing System and its effect on Environmental Predictions from Hours to a Season, Front. Mar. Sci., 6, 429, doi.org/10.3389/fmars.2019.00429, 2019.

---

## Author Response (AR2)

**Response to Topical Editor**

Sophie,

Thank you for having taken the time to review our paper and for your suggestions. You have a good eye for detail! I've made all the changes you suggest, and have reworded the sentence at line 276 regarding the tidal signal to improve readability.

A detailed response is provided below in blue.

Kind regards,

Greg

**Corrections from Topical editor**

- l.133: change "(See …" for "(see …"

done

- Title of 3.3: I would change "3.3 Modifications introduced for RIOPSv2" for "3.3 Modifications of SAM2 introduced for RIOPSv2"

done

- l.276: change "including tidal" for "including tides"

Changed the sentence to read "Satellite altimetry observations contain a variety of signals including those produced by tides"

- l.476: change "minimized" for "minimizes"

done

- l.569: YOPP should be defined the first time it appears in the text and not only at line 717

done